# Effects of drought and recovery on soil volatile organic compound fluxes in an experimental rainforest

Giovanni Pugliese [1,2] ✉, Johannes Ingrisch [1,3], Laura K. Meredith[4,5], Eva Y. Pfannerstill [2,11], Thomas Klüpfel[2], Kathiravan Meeran [3], Joseph Byron[2], Gemma Purser[6,7], Juliana Gil-Loaiza [4], Joost van Haren[4,5], Katerina Dontsova [5,8], Jürgen Kreuzwieser [1], S. Nemiah Ladd [1,9], Christiane Werner [1] & Jonathan Williams [2,10]

Drought can affect the capacity of soils to emit and consume biogenic volatile organic compounds (VOCs). Here we show the impact of prolonged drought followed by rewetting and recovery on soil VOC fluxes in an experimental rainforest. Under wet conditions the rainforest soil acts as a net VOC sink, in particular for isoprenoids, carbonyls and alcohols. The sink capacity progressively decreases during drought, and at soil moistures below ~19%, the soil becomes a source of several VOCs. Position specific [13]C-pyruvate labeling experiments reveal that soil microbes are responsible for the emissions and that the VOC production is higher during drought. Soil rewetting induces a rapid and short abiotic emission peak of carbonyl compounds, and a slow and long biotic emission peak of sulfur-containing compounds. Results show that, the extended drought periods predicted for tropical rainforest regions will strongly affect soil VOC fluxes thereby impacting atmospheric chemistry and climate.

Atmospheric biogenic volatile organic compounds (VOCs) are emitted and taken up by terrestrial ecosystems and play an important role in determining atmospheric processes that control air quality and climate. They react readily with the primary atmospheric oxidants hydroxyl radicals (OH) and ozone ($O_3$) leading to the production of secondary organic aerosol (SOA) particles. Such particles can in turn affect cloud formation, the earth's radiative balance and thereby climate[1]. Recent studies have shown that under certain condition the contribution of the soils to the total ecosystem VOC budget can even be comparable to that of the plants and they therefore represent an important component of the overall ecosystem VOC dynamic[2–4].

Soil VOC fluxes are inherently uncertain since they are the combined result of numerous biotic and abiotic processes. Biotic processes include microbial uptake, microbial decomposition of soil organic carbon (SOC) and plant residues, as well as emission from plant roots; while the abiotic processes include dissolution into or evaporation from soil water, adsorption onto or desorption from soil particles, reaction with soil chemicals, and evaporation from leaf litter[5–7]. Soil biotic and abiotic processes are in turn dependent on soil physio-chemical properties and environmental factors, e.g., temperature, soil pH, soil moisture, soil texture, soil porosity, nutrients, and SOC content. As a result, large variations in the composition,

[1]Ecosystem Physiology, Faculty of Environment and Natural Resources, University of Freiburg, Freiburg, Germany. [2]Atmospheric Chemistry Department, Max Planck Institute for Chemistry, Mainz, Germany. [3]Universität Innsbruck, Department of Ecology, Innsbruck, Austria. [4]School of Natural Resources and the Environment, University of Arizona, Tucson, AZ, USA. [5]Biosphere 2, University of Arizona, Oracle, AZ, USA. [6]UK Centre for Ecology & Hydrology, Penicuik, Edinburgh, UK. [7]School of Chemistry, The University of Edinburgh, Edinburgh, UK. [8]Department of Environmental Science, University of Arizona, Tucson, AZ, USA. [9]Department of Environmental Sciences, University of Basel, Basel, Switzerland. [10]Climate and Atmosphere Research Center, The Cyprus Institute, Nicosia, Cyprus. [11]Present address: Department of Environmental Science, Policy, and Management, University of California at Berkeley, Berkeley, CA, USA. ✉e-mail: g.pugliese@mpic.de

magnitude, and direction (i.e., emission vs uptake) of soil VOC fluxes have been observed as a function of ecosystem, season, diel dynamics, and environmental conditions[8–12].

The dependency of soil VOC fluxes on environmental conditions highlights the need to understand how soil VOC fluxes will change in response to human-accelerated climate change. Among the predicted impacts of climate change, temperature and drought frequency and duration are expected to increase worldwide[13]. Such events could potentially affect soil VOC fluxes, impacting the atmospheric budgets of certain VOCs with further and uncertain consequences on climate. This is particularly relevant for tropical rainforests as it is estimated that emissions from these ecosystems represent about 70% of the total source of biogenic VOCs to the atmosphere[14,15].

In this study, we conducted a long-term drought experiment (B2-WALD campaign[16]) in the enclosed experimental Biosphere 2 Tropical Rainforest (B2 TRF, Arizona, USA), to assess the effects of prolonged and severe drought followed by rewetting on soil VOC fluxes direction and magnitude. The soil VOC fluxes were monitored continuously and in real-time by means of a proton-transfer-reaction time-of-flight mass spectrometer (PTR-ToF-MS) connected to 12 closed dynamic soil chambers placed in 4 different sites of the B2 TRF. The ability to control and manipulate ecosystem conditions such as drought duration and the frequency of precipitation during rewet, makes the B2 TRF ideal for studying VOC dynamics[17]. Moreover, UV light filtering by the glass of the B2 TRF maintains low ozone concentration in the ambient air, precluding any complication of VOC signals through atmospheric oxidation chemistry. To identify the origin of emitted VOCs, we conducted additional tracer experiments with position-specific [13]C-labeled pyruvate both during pre-drought and during drought.

## Results
### Long term soil VOC fluxes dynamics
The drought-induced stress on the soil of the B2 TRF was monitored by means of soil moisture (volumetric water content) and soil matric potential (soil water availability to plants). Soil moisture (Fig. 1, purple

plot) and soil matric potential (Fig. 1, orange plot) decreased as the drought progressed, from 29 to 12.5% and from 0 to −3.7 MPa, respectively, and recovered back to pre-drought levels after the rain events. As expected, the soil temperature (Fig. 1, dark red plot) was relatively stable throughout the campaign (21.5–25.5 °C) and it showed a low diel variation with nighttime temperatures on average 1.4 ± 0.4 °C lower than the daytime temperatures. Part of this stability was achieved through the use of heaters to maintain nighttime air temperature above 15 °C during the latter part of the campaign in order to avoid having a seasonal trend in temperature which would have confounded the drought signal[16]. Soil respiration (Fig. 1, green plot) decreased similarly to the soil moisture, indicating that the soil microbial activity and root respiration decreased due to a reduction in soil water content[16,18].

Soil VOC fluxes showed distinct variations associated with the different periods of the campaign (Fig. 2). Generally, the B2 TRF soil acted as a net sink for all measured isoprenoid compounds namely isoprene, monoterpenes, and the isoprene oxidation products $C_5H_8O$ and methacrolein plus methyl vinyl ketone (MACR + MVK; both measured at m/z 71.049), with isoprene ($C_5H_8$) showing the highest net sink (Fig. 2, blue plots). Two types of temporal patterns were detected in the isoprenoids, namely the one displayed by isoprene and $C_5H_8O$ and the other by monoterpenes and MACR + MVK. Soil uptake of isoprene and $C_5H_8O$ peaked one week after the onset of the drought, then it steadily decreased with the development of the drought, but increased back to pre-drought levels during the rewetting period. Soil uptake of isoprene and $C_5H_8O$ can be attributed to the soil microbiome[7,17,19]. The fact that the soil uptake of these two VOCs decreased similarly to respiration in response to drought, can be seen as further evidence that the soil consumption of both VOCs was linked to the soil microbial activity. In contrast to isoprene and $C_5H_8O$, soil uptake of monoterpenes and MACR + MVK slightly increased during the drought. A possible explanation is that the microbes consuming isoprene/$C_5H_8O$ were more affected by drought than those consuming monoterpenes/MACR + MVK[16]. Another possible explanation would be that the

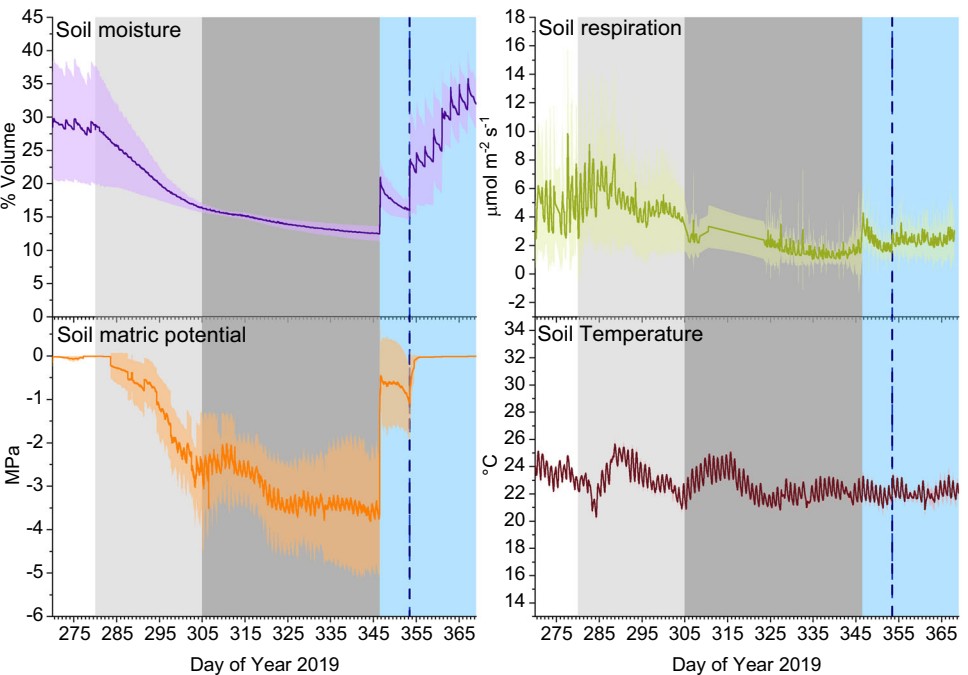

**Fig. 1 | Time series of soil moisture, soil matric potential, soil respiration, and soil temperature.** For soil respiration, line represents averaged value over the 12 soil chambers (*n* = 12) and the shaded area represents the standard deviation. For soil temperature, moisture and matric potential, lines represent averaged values over four sensors at 5 cm soil depth (*n* = 4) and the shaded areas indicate the standard deviation.

Background colors indicate the different phases of the campaign: pre-drought (white, Day of Year 270–279), early drought (light gray, Day of Year 280–305), severe drought (dark gray, Day of Year 305–346), and recovery (light blue, Day of Year 346–369). The first drought-ending rain event occurred at the start of the recovery period, and the vertical blue line indicates the time of the second rain event (Day of Year 353).

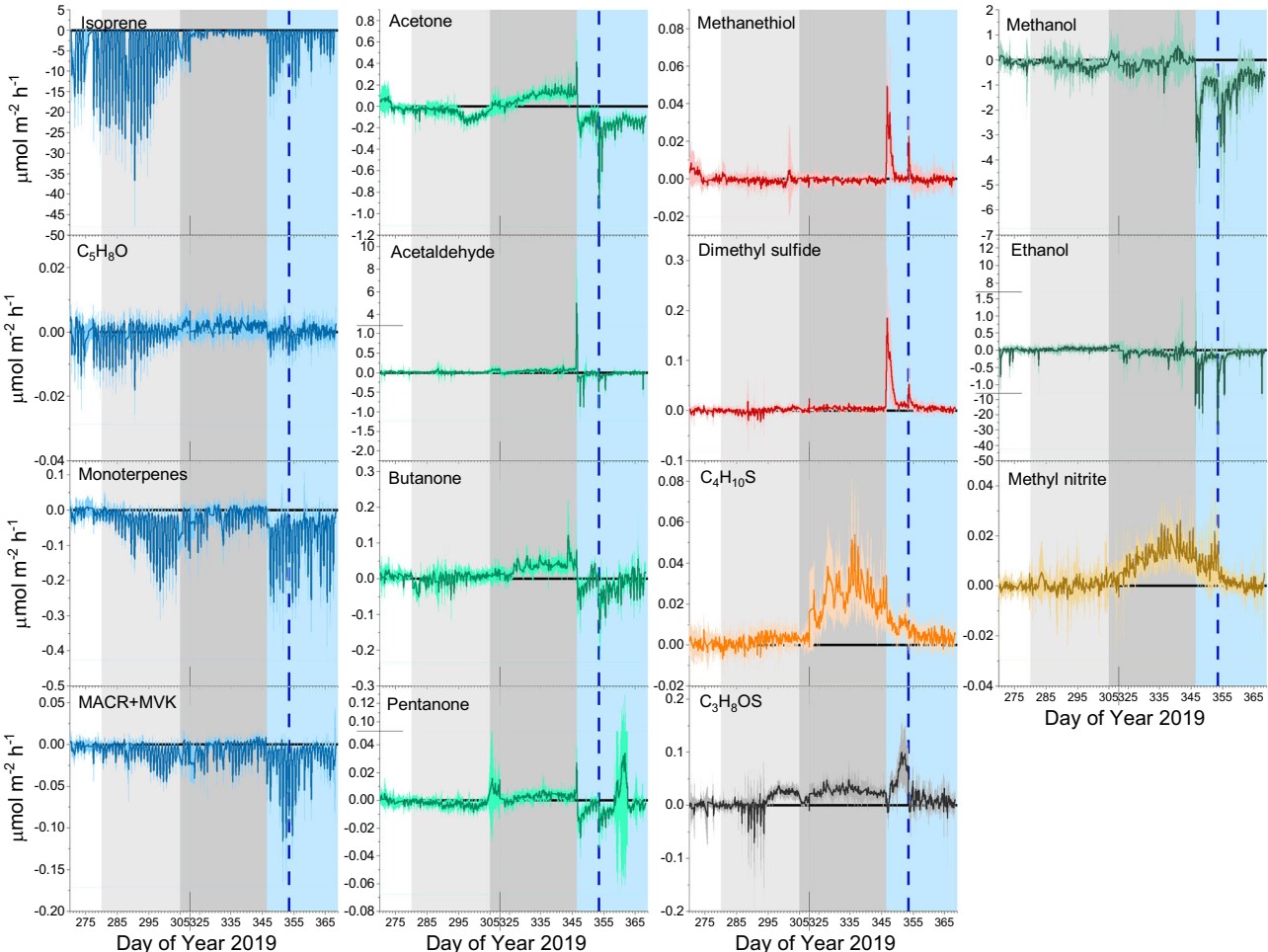

**Fig. 2 | Time series of soil volatile organic compound (VOC) fluxes.** Lines represent averaged fluxes over the 12 soil chambers (*n* = 12) and the shaded areas indicate the standard deviation. Background colors indicate the different phases of the campaign: pre-drought (white, Day of Year 270–279), early drought (light gray, Day of Year 280–305), severe drought (dark gray, Day of Year 305–346), and recovery (light blue, Day of Year 346–369). The first drought-ending rain event occurred at the start of the recovery period, and the vertical blue line indicates the time of the second rain event (Day of Year 353).

increased ambient air concentrations of monoterpenes and MACR + MVK (Fig. S1) during the drought may have induced an adaptation of the soil microbiome to consume these compounds at a relatively higher rate. Periodically, isolated soil flux measurements of speciated monoterpenes were made from manual soil chambers by GC-MS. These indicated that the uptake of the sum of monoterpenes measured by PTR-ToF-MS was mainly driven by (-)-α-pinene and (-)-β-pinene (Fig. S2), which were the monoterpene enantiomers most emitted by the vegetation in the B2 TRF[20]. In addition, GC-MS measurements showed that although the total monoterpenes fluxes resulted in a net uptake throughout the campaign, some monoterpenes such as terpinolene, γ-terpinolene, α-terpinene were mostly emitted by the soil. Soil emissions of monoterpenes have previously been observed in several ecosystems and have been attributed to microbial litter decomposition and to plant root emissions[21,22]. Considering the total monoterpenes flux, GC-MS and PTR-ToF-MS measurements showed a very similar trend and magnitude confirming the reliability of both measurement methods.

Soil fluxes of the carbonyl compounds acetone, acetaldehyde, butanone and pentanone all showed similar temporal variation patterns (Fig. 2, green plots), which differed from the aforementioned isoprenoid patterns. All carbonyls were taken up by the soil during the pre-drought period but as the drought progressed, the uptake gradually switched to emissions. Hence, the soil became a source of carbonyl compounds under severe drought. Apart from butanone, all

carbonyls showed a large emission peak immediately after the first rain event. Shortly after rewetting, all carbonyls shifted to be taken up by the soil again at rates about 10 times higher than pre-drought, but then the uptake steadily decreased to pre-drought levels. As observed for the isoprenoids, soil uptake capacity of all carbonyls during pre-drought and during the recovery period may be attributable to the higher microbial activity in wet soil compared to dry soil, although it cannot be excluded that their abiotic dissolution in the wet soil may have contributed to the total uptake. The gradual increase in soil emissions of all carbonyl compounds with increasing drought could be due to their production by soil microbes from soil organic matter as protective molecules in response to drought stress. Microbes under drought stress are known to generate high concentrations of chemicals internally to increase the osmotic potential of the cell and thereby draw more water from the surroundings[23].

Another distinct temporal pattern was identified for the two alcohols methanol and ethanol (Fig. 2, dark green plots) which were mostly taken up by the soil throughout the campaign. Their uptake slightly increased as the drought progressed and further increased after the two rain events. Soil uptake of methanol and ethanol during the whole campaign is also attributable to soil microbial activity. A wide range of soil microorganisms utilize ethanol and methanol as a carbon and energy source under both oxic and anoxic conditions[24,25] causing a net uptake in soil in diverse ecosystems and conditions[3,8,26,27]. The higher soil uptake capacity observed for both alcohols during the

recovery period could be due to the synergistic effect of an increased microbial activity, an increase abiotic dissolution in wet soil and increased ambient air concentrations (Fig. S1) attributable to higher plant leaf and litter emissions induced by rewet[28,29].

Soil fluxes were also detected for the sulfur containing compounds methanethiol and dimethyl sulfide (Fig. 2, red plots). They both showed low and highly variable soil fluxes for most of the campaign with a slight soil uptake for methanethiol and slight soil emission for dimethyl sulfide during severe drought. Both sulfur compounds showed two emission pulses directly after the rain events with the methanethiol emission pulse about one order of magnitude lower than dimethyl sulfide emission pulse. Methanethiol in soils originates mainly from the metabolism of sulfur-containing amino acids by microorganisms and from the methylation of hydrogen sulfide. Dimethyl sulfide is formed through the methylation of methanethiol, potentially explaining why the soil emission pulses observed for dimethyl sulfide were in general one order of magnitude higher than the emission pulse of methanethiol[30,31].The associated increase in ambient concentrations of both sulfur compounds (Fig. S1) after rain events clearly showed that soil can significantly contribute to local ambient concentrations of sulfur compounds.

The nitrogen containing compound methyl nitrite, $CH_3ONO$ (Fig. 2, brown plot) was also exchanged by the soil. This compound showed weak and highly variable fluxes both in and out of the soil during the pre-drought and early-drought periods. However, with the onset of severe drought methyl nitrite was consistently emitted and then during the recovery period the emission decreased steadily back to pre-drought levels. The observed methyl nitrite emission during severe drought is new and of potential significance to atmospheric chemistry in forests under drought stress as this molecule is readily photolyzed (lifetime ca. 2 min) to generate NO and formaldehyde[32]. The observed emission could be due to the reaction between nitrous acid (HONO) and lignin, which is one of the major constituents of soil organic matter[33,34]. Increased HONO emissions when soils dry out have been widely reported and associated with soil pH, chemical equilibrium with soil nitrite, heterogeneous hydrolysis of hydroxylamine, and to the release by soil ammonia-oxidizing microbes[35–37].

Soil emissions were also observed for other two sulfur-containing compounds $C_4H_{10}S$ and $C_3H_8OS$. $C_4H_{10}S$ showed a similar temporal pattern of methyl nitrite, with soil emissions progressively increased starting from the early-drought period reaching a maximum during severe drought and decreased back to pre-drought levels during the recovery period. In contrast, $C_3H_8OS$ soil emission started during early-drought and remained constant over the whole drought period. After the first rain event, $C_3H_8OS$ soil emission first suddenly decreased and then increased up to their maximum before the second rain event. After the second rain event $C_3H_8OS$ soil fluxes recovered to pre-drought levels. Soil emissions of $C_4H_{10}S$, tentatively identified as isopropyl methyl sulfide, have been previously reported but its origin is not well understood[38,39]. Mancuso et al.,[38] suggested $C_4H_{10}S$ as a potential intermediate product of the dimethyl sulfide metabolic pathway. However, in the present study, soil fluxes of dimethyl sulfide and $C_4H_{10}S$ showed rather different temporal patterns, indicating that they do not originate from the same soil process. We hypothesize that $C_4H_{10}S$ production in soils could either be due to microbial activity and, similarly to methyl nitrite, to secondary chemical reactions occurring on the soil surface. In contrast to $C_4H_{10}S$, the detection of $C_3H_8OS$ from soil is new and could be tentatively identified as 2-methylthioethanol, which is an intermediate product of methionine salvage pathway by microbes[40]. As methionine production demands high energy[40], water stressed soil microbes recycled it leading to higher 2-methylthioethanol soil emissions.

It should be noted that soil VOC fluxes are both a function of soil processes and ambient concentrations above the soil (Fig. S1). For VOCs that were both emitted and taken up by the soil, namely $C_5H_8O$,

acetone, acetaldehyde, butanone, and pentanone, no relationship was found between soil fluxes and respective ambient concentrations. Thus, no fixed compensation point, i.e., the ambient VOC concentration at which the soil flux is zero, could be identified. This is because the soil fluxes for these VOCs were mainly driven by soil moisture levels, which dramatically changed over the campaign, masking the influence of the ambient concentrations. Nevertheless, to assess the effect of the VOC concentrations in the ambient air on VOC soil uptake rates, we calculated the deposition velocities, which are defined as the ratio of VOC uptake rates to their ambient concentrations. The trends in deposition velocities (Fig. S3) were very similar to those of the net uptake fluxes (Fig. 2). However, during the recovery period, net soil uptake rates of isoprene were lower compared to pre-drought period (Fig. 2), while isoprene soil deposition velocity had actually returned to pre-drought levels (Fig. S3). This indicates that the lower net isoprene soil uptake during the recovery period compared to the pre-drought period was due to a lower isoprene concentration in the ambient air (Fig. S1) and not to a reduction in microbial uptake capacity.

Soil moisture was a key driver for VOC fluxes and the relationship between the average of normalized VOC fluxes and soil moisture was non-linear evolving around a soil moisture threshold of ~19%, as determined by segmented regression (Fig. 3). Below this soil moisture threshold, the VOC uptake capacity of the soil dramatically decreased and the soil started to be a source of VOCs. This suggests that 19% represents the soil moisture threshold corresponding to the point when the water-stressed soil microbes started producing and accumulating protective osmolytes, including VOCs, to reduce their internal water potential to avoid dehydrating and dying[23,41].

## Rewetting dynamics

Temporal dynamics of soil respiration and soil VOC fluxes following soil rewetting events were analyzed in detail (Fig. 4). To capture fast soil flux rewet dynamics, 3 soil chambers placed on 3 different sites of the B2 TRF were manually rewetted at 5:30 am on 12th December 2019 and for the following 5 h were measured with a high frequency, i.e., each chamber was measured every 30 min (Fig. 4a). At 11 am on 12th December 2019, the remaining 9 chambers were subjected to the first whole forest rain rewet (Fig. 4b) involving simulated rainfall from the roof mounted sprinkler system, while the 3 manually rewetted chambers were covered with rainout shelters. From 30 min before the whole forest rewet, all the 12 chambers were measured consecutively with a temporal resolution of 2 h. A second rainfall event on the whole forest involving all 12 soil chambers was conducted one week later on 19th December 2019 at 11:00 (Fig. 4c).

A pulse in $CO_2$ soil emissions was observed after all rewet events (Fig. 4, purple plots). This phenomenon is known as the "Birch effect" and has been attributed to a rewetting-induced mineralization of labile soil organic carbon pools[42,43]. The increased availability of these organic substrates after the rewet is thought to be due to an increased release of intracellular osmolytes accumulated by water-stressed microorganisms, to microbial cell lysis caused by osmotic shock, and to the physical disruption of soil aggregates protecting organic matter[23,44,45]. An emission pulse was also observed for carbonyl compounds namely acetaldehyde, acetone, butanone and pentanone (Fig. 4, light green plots) after the rewet events on 12th December, and for the sulfur compounds, namely methanethiol and dimethyl sulfide (Fig. 4, red plots) after all rewet events. The carbonyls pulse was fast and short as it occurred within 2–4 h after the rewets and it lasted for about 2 h. This indicates that the carbonyl pulse was abiotic in origin and attributable to the immediate water-induced mobilization of the soil organic carbon such as the rapid release of cell osmolytes, cell lysis caused by osmotic shock, and physical disruption of the soil organic matter[44,45]. In addition, as the production and emission of carbonyls by the soil increased during drought (Fig. 2), the soil micropores could have been filled with these compounds, and when rain water entered

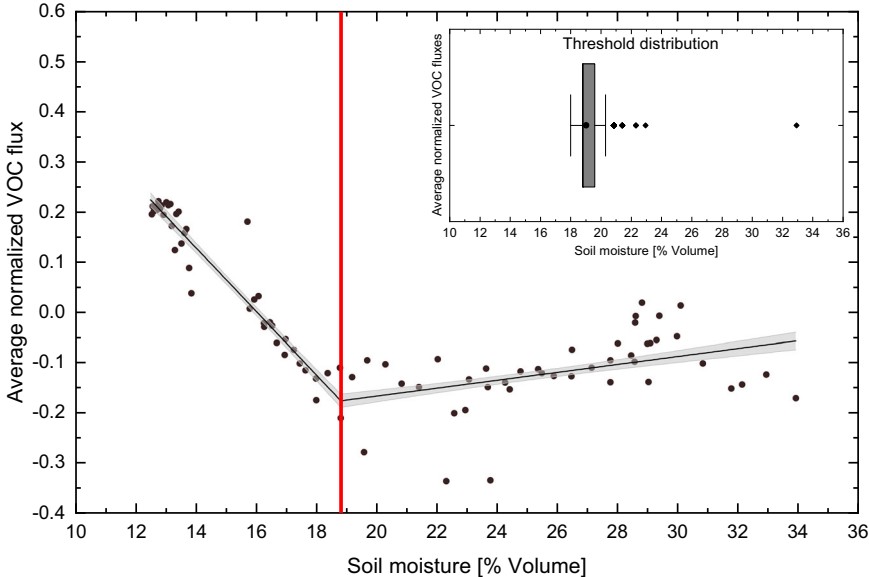

**Fig. 3 | Relationship between soil volatile organic compound (VOC) fluxes and soil moisture.** Segmented regression analysis between the averaged normalized soil VOC fluxes ($n = 15$ VOCs) and soil moisture, over the whole period of the campaign. To give the same weight to all VOC fluxes, for each VOC, daily averaged soil fluxes were normalized to their absolute maximum. Dots represents the average of normalized fluxes over all VOCs shown in Fig. 2, for each day of the campaign. Black lines indicate the segmented regression model with the moisture threshold indicated by the vertical red line. Shaded gray area indicate the 95% confidence interval of the regression model. The box plot chart shows the distribution of threshold estimates based on 1000 bootstrapped samples. The box represents 25% to 75% of the dataset. The filled circle and central line indicate the mean and median value, respectively. The whiskers indicate the minimum and maximum data points at 1.5 times the interquartile range. Filled diamond indicate the outliers.

the micropores the water molecules replaced those of the carbonyls causing their release into the ambient air. In contrast to the carbonyls, the pulse of the sulfur compounds was slow (occurred 9 h after the manual and first rain rewet and 4 h after the second rain rewet) and long (it lasted for about 3 days), and it was concurrent with and very similar to the soil respiration pulse. These are strong indicators of the biotic origin of the sulfur pulse, attributable to a water induced mobilization and mineralization of the soil organic sulfur pools, whereby large insoluble sulfur-containing organic molecules are reduced to smaller soluble sulfur containing molecules by soil microbes or by extracellular soil enzyme[46,47]. As shown in Fig. 1 for $CO_2$ and in Fig. 2 for methanethiol and dimethyl sulfide, the emission pulses following the second rain event were significantly lower in absolute magnitude compared to the pulses following the first rain rewet, indicating that shorter drought-rewetting cycles induce a lower mobilization and mineralization of the soil organic matter or that induce a lower build-up of substrate pools[48,49]. Indeed, the subsequent rain events conducted every second day starting from 21 December did not induce any VOCs and $CO_2$ soil emission pulses.

The soil uptake rates of isoprene, $C_5H_8O$ and monoterpenes increased considerably only the day after the rewets (Fig. 4, blue plots), reflecting the time needed for the microbes responsible for the consumption of these compounds to completely restore their activity. In contrast, the uptake of MACR + MVK peaked within a few hours after the rewet events most probably due to its abiotic dissolution in wet soil. An increase in soil uptake of alcohols was observed within 4 h after the rewets (Fig. 4, dark green plots) as a consequence of the simultaneous increase in their ambient concentrations (Fig. S1) and microbial activity, and to their abiotic dissolution in wet soil. In response to all rewet events, $C_3H_8OS$ soil emission considerably decreased and the soil switched to taking up $C_3H_8OS$, but after a few hours the emission was restored again. Methyl nitrite emission (Fig. 4, brown plots) slowly decreased in response to the rewet events likely as a consequence of decreasing HONO production with increasing soil moisture[36]. $C_4H_{10}S$ fluxes decreased in response to the rewets in a similar fashion of

methyl nitrite suggesting that the soil emissions of these two compounds could have originated from similar processes.

To examine whether the soil physicochemical properties measured at four different sites of the B2 TRF (Table 1) contributed to the soil VOC fluxes over the rain rewet, partial least square regression (PLSR) analysis was conducted. In addition to the soil physicochemical properties reported in Table 1, soil moisture (volumetric water content), soil matric potential (soil water availability to plants) and soil respiration measured from the four sites were also included as predictors in the PLSR analysis. For each site, averaged soil VOC fluxes over the last two days of drought and over the first seven hours of rewet were used for PLSR analysis. The soil fluxes of VOCs from the same compound class and that showed similar rewetting dynamics were averaged. Therefore, isoprenoid compounds include isoprene, $C_5H_8O$, MACR + MVK, and monoterpenes; carbonyl compounds include acetaldehyde, acetone, butanone, and pentanone; alcohol compounds include methanol and ethanol; sulfur compounds include methanethiol and dimethyl sulfide; soil fluxes of methyl nitrite, $C_4H_{10}S$, and $C_3H_8OS$ were considered individually. Soil fluxes for each individual VOC from the four sites of the B2 TRF over the last two days of severe drought and over the first seven hours of the rain rewet are shown in Fig. S4. In Fig. 5a–g regression coefficients and variable of importance (VIP) are shown from each individual PLSR analysis. Soil moisture, soil matric potential, soil respiration, and soil clay content were the most important variables (VIP > 1) for the most of the VOCs. Higher soil water content during the rain rewet, induced a higher release in the ambient air of carbonyl compounds (Fig. 5b) accumulated in the soil micropores and, at the same time, a higher abiotic dissolution alcohol compounds (Fig. 5c) from the ambient air. Additionally, increased soil water content after rewet also increased the microbial activity, leading to a higher soil uptake of isoprenoids (Fig. 5a) and alcohol compounds (Fig. 5c), and to a higher soil emission of sulfur compounds (Fig. 5d). In contrast, soil emissions of methyl nitrite (Fig. 5e) and $C_3H_8OS$ (Fig. 5f) negatively correlated with soil respiration indicating that soil emissions of these compounds

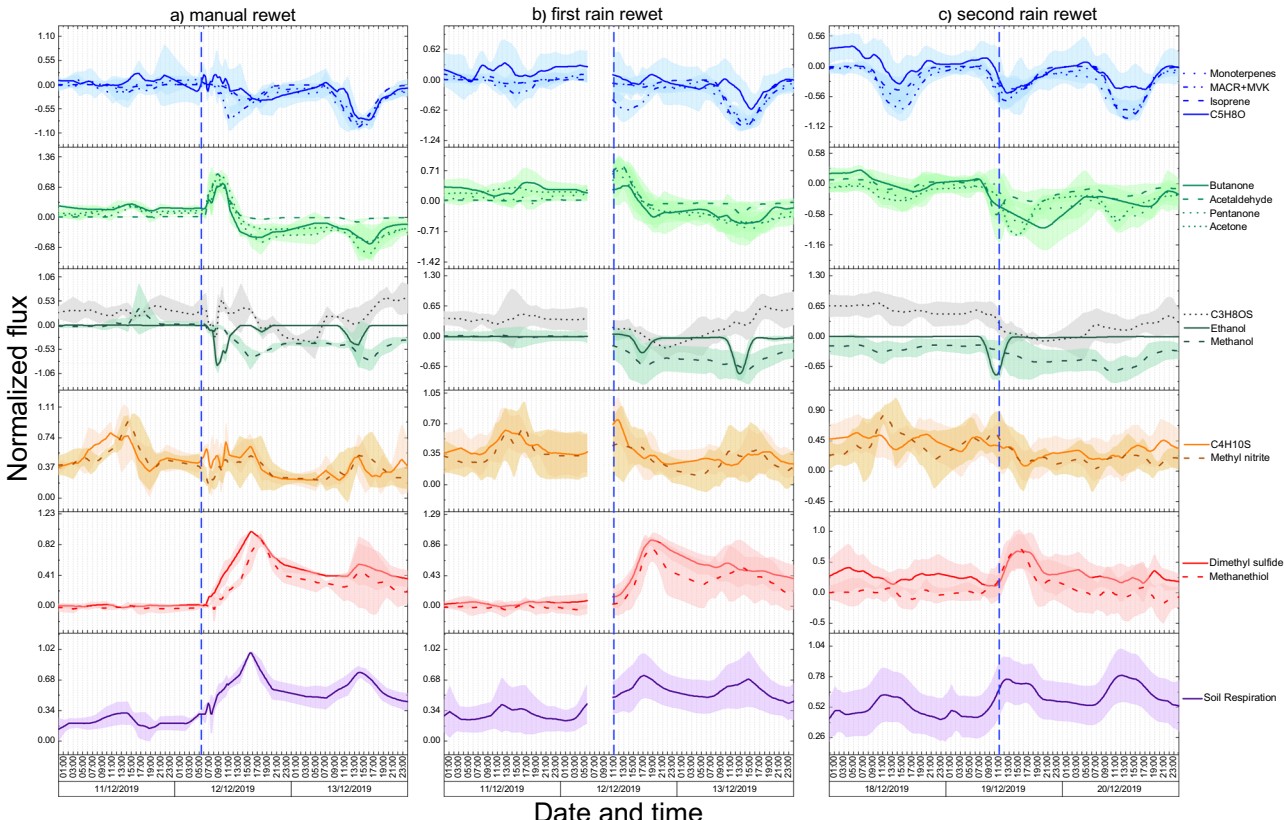

**Fig. 4 | Temporal dynamics of soil fluxes following soil rewetting.** Soil respiration and soil volatile organic compound (VOC) fluxes after **a** manual rewet, **b** first rain rewet, **c** second rain rewet. To make comparable the dynamics of different compounds, for each compound, soil fluxes were normalized to their respective absolute maximum. Lines represents averaged values over 3 chambers for manual rewet ($n = 3$), over 9 chambers for first rain rewet ($n = 9$), and over all 12 chambers for second rain rewet ($n = 12$). The shaded areas indicate the standard deviation. The data gap on the first rain rewet plots is due to the fact that during that time only the manual rewetted chambers were measured with a high temporal resolution with the aim to capture the fast dynamics.

decreased with increasing soil microbial activity. Soil clay content negatively correlated with all soil VOC fluxes. The effect of soil clay content was interconnected to the soil water content. Higher soil water content was associated with lower VOC sorption capacity of the clay minerals due to its hydrophilic character[5]. Moreover, when clay minerals are wetted, they can swell, decreasing porosity, VOC diffusion, and therefore both VOC emission and uptake. This explains the negative effect of soil clay content on soil VOC emissions and uptakes over the rain rewet.

## Diel dynamics

Diel dynamics of soil VOC fluxes, deposition velocities and environmental variables were analyzed for each period of the campaign (Fig. 6). As exemplified for isoprene (Fig. 6a), soil fluxes of all isoprenoids showed a daytime maximum and minimum uptake rates at night, closely following the diel cycle of their ambient concentrations (Fig. 6b). A similar diel cycle was observed during the recovery period for acetone, acetaldehyde, butanone, pentanone and methanol, all of which were taken up by the soil (Fig. S5). In contrast, the soil emissions of carbonyl compounds as well as those of dimethyl sulfide and $C_3H_8OS$ did not show any diel cycle. A diel cycle was also observed for isoprenoid deposition velocities (Fig. 6c) during all periods of the campaign, and for methyl nitrite emissions (Fig. 6d) and $C_4H_{10}S$ emissions (Fig. 6e) only during the severe drought period.

To identify potential process drivers, a correlation analysis of the average hourly values was performed between environmental factors that followed a diel cycle (i.e., soil temperature (Fig. 6g) and soil matric potential (Fig. 6h)) and soil VOCs fluxes and deposition velocities. However, no correlation was found, indicating that the observed diel

cycles of VOC fluxes were not driven by these environmental factors. The decrease observed in isoprenoid deposition velocity during the night was most probably due to substrate limitation in a very depleted ambient air at night as shown in Fig. 5b. This is because plant emissions of isoprenoids are metabolically linked to plant photosynthesis with higher emission at higher photosynthetic rates[20]. As a result, ambient concentrations of isoprenoids are during nighttime are low. Higher daytime emission of methyl nitrite compared to nighttime can be attributed to a higher HONO production during the day which has been attributed to nitrogen dioxide ($NO_2$) reduction on light activated humic acids of the soil organic matter[50]. Although the UV light is filtered by the B2 TRF glass, the $NO_2$ reduction to HONO is also very effective under visible light[50]. The diel cycle also observed for $C_4H_{10}S$ soil emissions support the hypothesis that it could have originated from soil surface light dependent chemical reactions, similarly to methyl nitrite.

## Origin of VOC emissions

To identify the origin of the emitted VOCs, the soil was labeled with position specific $^{13}C_1$-pyruvate and $^{13}C_2$-pyruvate. A net soil emission was observed for the flux of the fractional abundance of $^{13}C$-acetone (defined as $^{13}C$-VOC/($^{13}C$-VOC + $^{12}C$-VOC)), after $^{13}C_2$-pyruvate injections both during pre-drought and during drought period (Fig. 7). Absolute fluxes of $^{13}C$-acetone are shown in Fig. S6. This is clear evidence that soil microbes are able to produce VOCs from precursors in the soil and that the emissions observed were not just due to abiotic release from soil. As shown in Fig. 2, acetone was mainly consumed under wet soil conditions; therefore, the emission observed for the fractional abundance of $^{13}C$-acetone during pre-drought demonstrated that soil

**Table 1 | Soil physiochemical properties at each site of the B2 TRF where soil flux chambers were placed**

| Site | S1 | S2 | S3 | S4 |
|------|-----|-----|-----|-----|
| Texture | Loam | Loam | Loam | Silt Loam |
| % Clay, <2 µm | 17.90 ± 1.74 | 23.75 ± 0.00 | 17.93 ± 0.03 | 25.07 ± 0.19 |
| % Silt, 2–50 µm | 46.13 ± 3.41 | 36.25 ± 0.00 | 44.80 ± 0.15 | 51.37 ± 0.73 |
| % Sand, 50–2000 µm | 35.99 ± 4.96 | 39.24 ± 2.88 | 37.27 ± 0.18 | 23.57 ± 0.91 |
| % Gravel | 19.79 | 15.12 | 26.36 | 19.06 |
| Bulk Density, g/cm$^3$ | 1.38 ± 0.20 | 1.40 ± 0.21 | 1.21 ± 0.29 | 1.40 ± 0.15 |
| Porosity | 0.48 ± 0.07 | 0.47 ± 0.08 | 0.54 ± 0.11 | 0.47 ± 0.08 |
| Total Carbon, µg/mg | 24.10 ± 8.02 | 27.00 ± 3.06 | 29.40 ± 7.30 | 21.44 ± 6.51 |
| Total Nitrogen, µg/mg | 1.90 ± 0.69 | 2.26 ± 0.18 | 2.54 ± 0.80 | 1.70 ± 0.42 |
| Electrical conductivity, µS/cm | 409.10 ± 130.7 | 771.30 ± 102.4 | 644.00 ± 277.90 | 257.10 ± 77.80 |
| pH | 7.26 ± 0.08 | 7.44 ± 0.14 | 7.31 ± 0.11 | 6.98 ± 0.32 |
| Water holding capacity, % | 66.94 ± 6.70 | 60.48 ± 3.38 | 59.03 ± 2.82 | 70.78 ± 4.85 |

microbes can both produce and consume acetone and that under wet conditions they were able to consume more acetone than they actually produced as discussed in Honeker et al., 2023[51]. During drought the emission fluxes of the fractional abundance of $^{13}$C-acetone was about one order of magnitude higher than during pre-drought. This is further evidence that under drought stress soil microbes used energy resources to support higher VOC production[23,51].

## Discussion

In normal wet conditions, the soil of the experimental rainforest acted as a net VOC sink. The soil uptake capacity progressively decreased in response to increasing drought and, under severe drought conditions the soil started to be a source of several VOCs, including carbonyls and methyl nitrite, $C_4H_{10}S$ and $C_3H_8OS$. This trend could be attributable to the soil microbes that significantly reduced their consumption of atmospheric VOCs under drought stress, and to prevent osmotic imbalance, relocated carbon and nitrogen resources from growth pathways to produce and accumulate osmolytes, including VOCs[23,51]. This was further confirmed by the position specific $^{13}$C-pyruvate experiments that clearly demonstrated that soil microbes can be a significant source of VOCs from available energy sources and that their energetic investment in VOCs production was higher during drought[51]. The moisture threshold below which the soil microbes dramatically reduced consumption of atmospheric VOCs and started to become a source of several VOCs was 19%. Currently the soil moisture conditions normally experienced in tropical rainforests such as the Amazon are higher. However, continued global warming, deforestation, and the predicted increased frequency of El Niño events is likely to reduce soil moisture levels to below this threshold for longer periods inducing associated VOC emission events[13]. For instance, during the strong El Niño drought in 2015/2016 a negative soil moisture anomaly with an average reduction of almost 30% was reported in the Amazon basin and at same time large pulses of unexplained OH reactivity were observed in the same region[52,53]. Reduced soil VOC uptake capacity as well as increased soil and plant VOC emissions induced by the El Niño drought could represent a potential explanation for the OH reactivity pulse[53]. Therefore, the results shown in the present study have implications to near future real-world scenarios. Increased soil VOC emissions in combination with a reduction of the atmospheric VOCs uptake by the soil will affect the rainforest ecosystem atmospheric chemistry with possible further feedback on radiative effects and climate. These emission effects will be further exacerbated by soil rewetting events following prolonged drought periods as shown by the large pulse in soil VOC and $CO_2$ emissions observed after the rainfall events. The rewetting induced a rapid and short abiotic emission peak of carbonyl compounds attributable to water-induced release of cellular osmolytes, cell lysis caused by osmotic shock, water-induced physical degradation of soil organic matter, and to purging of the carbonyls accumulated in the soil micropores induced by the rain water entering the soil micropores. The subsequent microbial mineralization of these mobilized carbon sources resulted in a $CO_2$ pulse (Birch effect)[44]. The emission pulse observed for the two sulfur compounds namely methanethiol and dimethyl sulfide after soil rewetting was similar to the Birch effect and was therefore attributable to the mineralization of the soil organic sulfur pools by soil microbes and enzymes. The increase also observed in ambient concentrations occurred simultaneously to the soil emission pulses of the two sulfur compounds, showing that soil can significantly contribute to local ambient concentrations of sulfur compounds. It should be noted that on the global scale, the ocean is a larger source of dimethyl sulfide than the rainforest[54]. However, due to the relatively short atmospheric lifetime of dimethyl sulfide in the tropics (ca. 1 day) and the stronger convection experienced overland, rainforest emissions can still be important to local and regional chemistry. Dimethyl sulfide is of high relevance in atmospheric chemistry as it can be oxidized to sulfuric acid, contribute to new particle formation, and ultimately grow to form cloud condensation nuclei[55,56]. The emission of the additional sulfur-containing compound, i.e., methanethiol, would strengthen this effect.

Soil clay content played an important role in determining soil VOC fluxes over the rain rewet. Soil VOC uptake and emission rates both decreased with increasing soil clay content. This is because the capacity of the clay minerals to sorb and release VOCs decreased with increasing soil water content due to their hydrophilic character and to their tendency to swell in wet conditions[5].

Soil fluxes of several VOCs followed a diel cycle with higher emission and uptake rates both occurring during daytime compared to nighttime. Soil uptake rates of isoprenoids closely followed the diel cycle of their atmospheric concentrations, while diel cycles in methyl nitrite and $C_4H_{10}S$ emissions were a consequence of light dependent processes at the soil surface. Methyl nitrite emissions are highly relevant for the atmospheric chemistry in the boundary layer due to its rapid photolysis to NO and formaldehyde[32]. Both compounds serve to elevate concentrations of OH radicals, the atmosphere's primary oxidant. Since methyl nitrite emissions occurred only during drought, the presented results suggest that future climate change associated drought periods will also affect diel rainforest carbon cycles and atmospheric oxidation chemistry.

Prolonged drought and recovery had a major impact on soil VOC fluxes in the experimental rainforest, affecting the composition and quantity of VOCs in the atmosphere of the enclosed ecosystem[16]. Soil VOC fluxes and their parameterization related to soil moisture levels must be included in atmospheric models to simulate current atmospheric chemistry and to improve climate model predictions of ecosystem responses to drought.

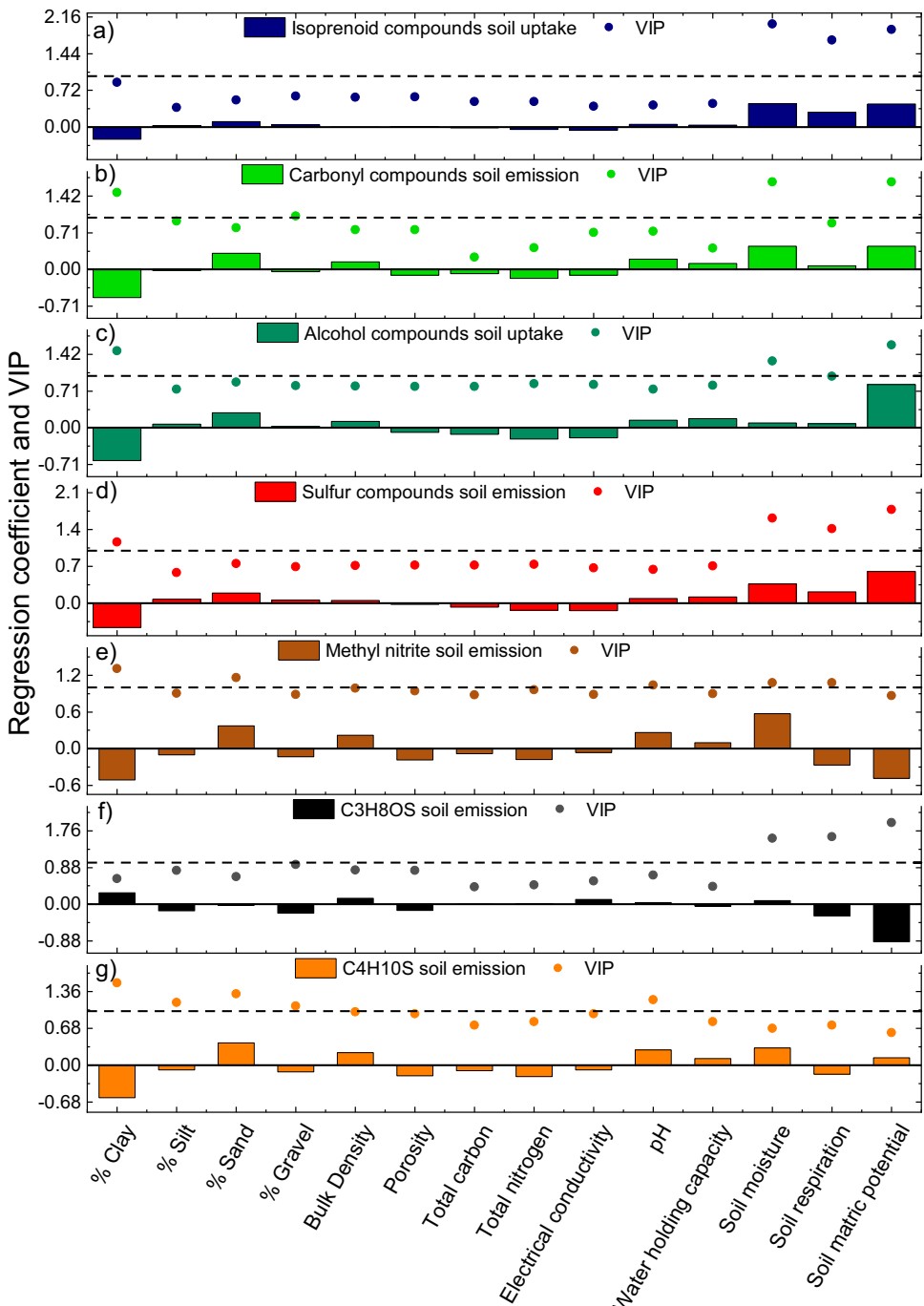

**Fig. 5 | Relationship between soil properties and soil volatile organic compound (VOC) fluxes over the rain rewet.** Regression coefficients and Variable Importance (VIP) from partial least squares regression (PLSR) analysis between the measured soil variables at four sites of the Biosphere 2 Tropical Rainforest (B2TRF) and **a** isoprenoid compounds soil uptake, **b** carbonyl compounds soil emission, **c** alcohol compounds soil uptake, **d** sulfur compounds soil emission, **e** methyl nitrite soil emission, **f** C$_3$H$_8$OS soil emission, **g** C$_4$H$_{10}$S soil emission. For isoprenoid compounds, soil fluxes of isoprene, C$_5$H$_8$O, MACR + MVK (methacrolein plus

methyl vinyl ketone), and monoterpenes were averaged; for carbonyl compounds, soil fluxes of acetaldehyde, acetone, butanone, and pentanone were averaged; for alcohol compounds, soil fluxes of methanol, and ethanol were averaged; for sulfur compounds, soil fluxes of methanethiol, and dimethyl sulfide were averaged. Positive regression coefficients indicate a positive relationship and negative ones a negative relationship. Variables with a VIP > 1 are considered important while variables with VIP < 1 are considered less important.

## Methods

### The B2 TRF mesocosm and controlled drought experiment

The B2 TRF mesocosm is a fully enclosed ecosystem which allows temperature, humidity, atmospheric gas composition, and precipitation to be manipulated. The B2 TRF mesocosm demonstrated similar behavior to the world's tropical rainforests and it allowed the study tropical ecosystem responses to environmental changes[17,57]. The

mesocosm has an area of 1940 m$^2$ and a volume of 26700 m$^3$ and the vegetation is rooted in 2–4 m of soil. The low ozone (O$_3$) concentration (ca. 1 ppbV) and the reduced hydroxyl radicals (OH) formation inside the B2 TRF due the UV-light filtering by the glass, prevent VOCs oxidation allowing the estimation of the fluxes of highly reactive VOCs since the ambient VOC concentrations reflect the ecosystem VOC dynamics. The soil flux measurements were conducted during the

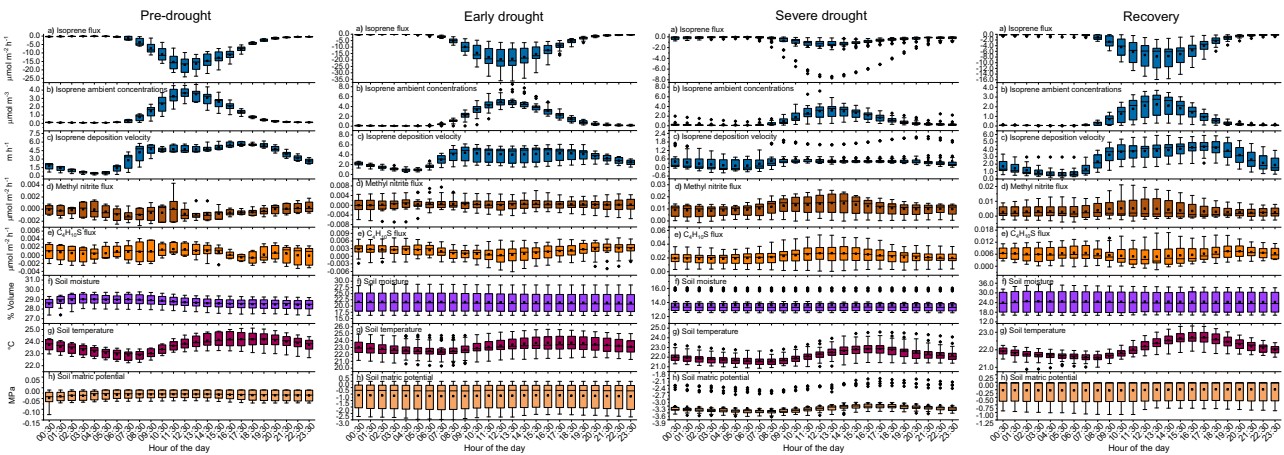

**Fig. 6 | Diel cycle during the different period of the campaign.** Diel cycle observed during pre-drought (*n* = 10 days), early drought (*n* = 26 days), severe drought (*n* = 27 days, the days of $^{13}C$-pyruvate experiments were excluded), and recovery (*n* = 21 days) for **a** isoprene soil flux, **b** isoprene ambient concentration, **c** isoprene deposition velocity, **d** methyl nitrite flux, **e** $C_4H_{10}S$ flux, **f** soil moisture, **g** temperature, and **h** soil matric potential. The boxes represent 25% to 75% of the dataset. The filled square and central lines indicate the mean and median values, respectively. The whiskers indicate the minimum and maximum data points at 1.5 times the interquartile range. Filled diamond indicate the outliers.

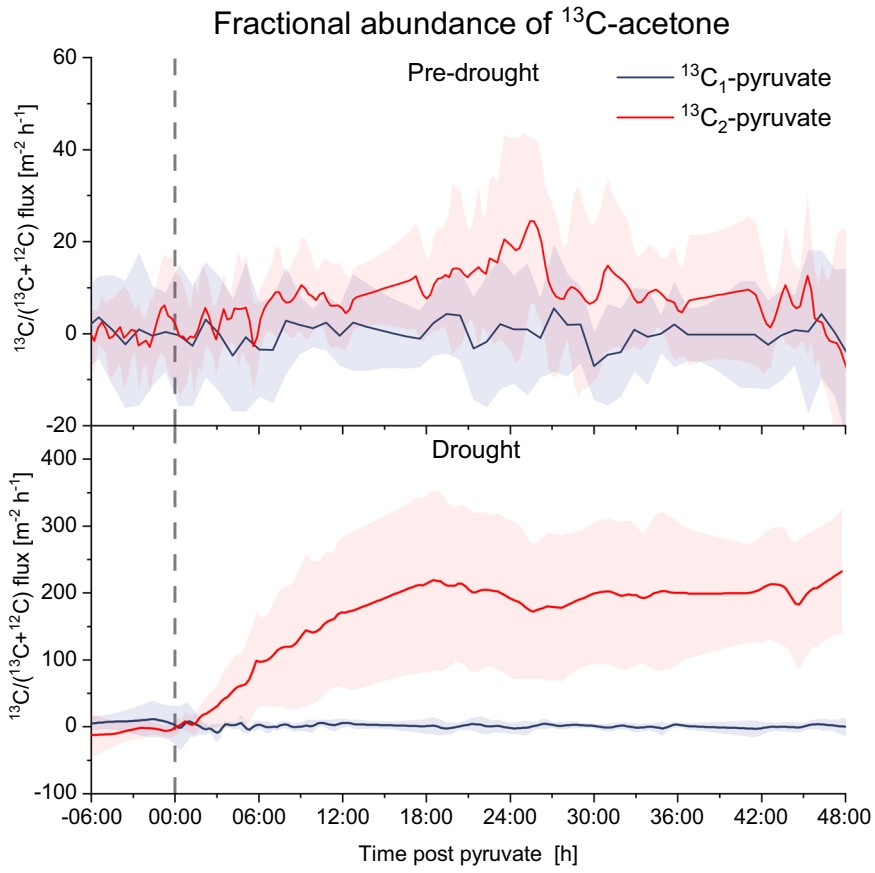

**Fig. 7 | Soil emissions of fractional abundance of $^{13}C$-acetone from 6 h prior to 48 h after $^{13}C$-pyruvate soil injections.** Soil emission fluxes of the fractional abundance of $^{13}C$-acetone (defined as $^{13}C$-VOC/($^{13}C$-VOC + $^{12}C$-VOC)) after the C$_1$-$^{13}C$-pyruvate (blue lines) and C$_2$-$^{13}C$-pyruvate (red lines) soil injections during pre-drought (upper panel) and drought (lower panel) period. Lines represent averaged fluxes over the 9 chambers. The shaded areas indicate the standard deviation.

Water, Atmosphere, and Life Dynamics campaign (B2-WALD)[16] from September 2019 to January 2020 during which 65 days of drought were induced. During the whole campaign, the temperature inside the ecosystem was controlled in order to avoid having a seasonal trend in temperature which would have confounded the drought signal[16]. The campaign started with a pre-drought phase, with rainfall regime kept at a rate of about 30 mm per week. The drought phase started after the last rainfall event at midnight 7th October 2019. From 1st November 2019 to 2nd December 2019 the relative humidity of the ecosystem was further reduced enhancing the drought conditions. After 65 days

of drought, at 5:30 am on 12th December 2019, 3 soil chambers placed on 3 different sites of the B2 TRF were manually rewet by adding ~2.2 L (~22.5 mm) of water per chamber. Soil fluxes from these chambers were measured with a high frequency, i.e., each chamber was measured every 30 min for 5 h, with the aim to capture fast soil flux rewet dynamics. Rainout shelters were placed over the 3 manually rewetted chambers while, at 11 am on 12th December 2019, the remaining 9 chambers were subjected to the whole forest rewet. From 30 min before the whole forest rewet, all the 12 chambers were measured consecutively with a temporal resolution of 2 h. Precipitation was delivered via overhead sprinklers and ~35000 L (~18 mm) of water were added over 4.5 h period. A second rainfall event on the whole forest involving all 12 soil chambers was conducted one week later on 19th December 2019 at 11:00, by adding ~ 36000 L (~19 mm) of water in 4.5 h. Subsequent rain events were then conducted every second day starting from midnight on 21st December 2019.

## Experimental set-up

Soil VOC fluxes were measured continuously using a proton transfer time of flight mass spectrometer (PTR-ToF-MS-8000, Ionicon Analytik GmbH, Innsbruck, Austria) directly connected to the outflow of an automated soil flux measuring system consisting of a LI-8100 infrared gas analyzer (IRGA; for $CO_2$ fluxes measurement), a LI-8150 16-port multiplexer (Licor Inc., Lincoln, NA, USA) and 12 dynamic soil flux chambers (LI 8100-104 Long-Term Chambers with opaque lids, Licor Inc.). The soil flux measurement is based on a closed dynamic system. The ambient air was first sampled during the pre-purge period (2.5 min) with chamber lid open, and then once the chamber lid was closed, the enclosed air was recirculated inside the system during the flux measurement period (6.5 min). The soil flux measurement is based on the calculation of the compound concentration development in the recirculated ambient air. The soil chambers are featured with a pressure vent system to maintain pressure equilibrium inside the chambers even under windy conditions[58].

The working principle of the PTR-ToF-MS instrument is based on a soft ionization process which consists of a proton transfer from hydronium ions ($H_3O^+$) to sample VOCs having a higher proton affinity than water (691 kJ mol$^{-1}$). Protonated VOCs are then analyzed in a high-resolution time-of-flight mass spectrometer according to mass-to-charge ratio (m/z)[59]. The instrumental settings were as follows. The PTR drift tube pressure was 2.2 mbar, the PTR drift tube voltage was 600 V, and the PTR drift tube temperature was 60 °C, resulting in an E/N ratio of 137 Td. The time resolution was 10 s with the m/z monitored up to 500 Da.

The total volume of the soil flux system, including chamber, tubing, IRGA and multiplexer, was about 6.5–7 L. For gas analysis, ca. 100 sccm were subsampled via a T-piece placed at outflow of the LI-8100A and were distributed to the different analyzers including the PTR for VOCs measurement. To avoid negative pressure, ca. 100 sccm of synthetic air were introduced in the soil flux system via a T-piece placed at the inflow of the Li-8100A. In order to minimize surface effects on VOC analysis, perfluoroalkoxy (PFA) tubing was used for the soil flux system, for the subsampling line and for the PTR inlet[60]. The PTR sampling flow was 30 sccm and the PTR inlet temperature was 60 °C. All PTR-TOF files were processed using the software PTRwid (version v003)[61]. The ion yields of all m/z were measured in counts per second (cps) and compounds were identified from the measured exact m/z of their protonated parent ions and isotopic patterns. To account for possible variations of the reagent ion signals, measured ion intensities were normalized to the $H_3O^+$ counts in combination with the water-cluster ion counts[62]. Only compounds with signal intensities higher than the instrumental background were considered for further analysis.

Nocturnal calibrations, starting from midnight, were performed using a standard gas cylinder containing different multi-VOC

component calibration mixtures in Ultra-High Purity (UHP) nitrogen (Apel-Riemer Environmental, Inc., Colorado, USA). Two calibration standard cylinders were used during the campaign to allow explicit calibration of a wide range of VOCs. The first cylinder was used for two periods: from 18th September 2019 to 6th November 2019; and from 17th December 2019 to 20th January 2020. The second cylinder was used from 7th November 2019 to 16th December 2019. VOC gas standards included in the two calibration standard cylinders with their respective detection limit (LOD) and total uncertainty are reported in Table S1. For daily calibration the VOC mixture was subjected to 5-step dynamic dilutions by means of a liquid calibration unit (LCU, IONICON Analytik, Innsbruck, Austria). The gas standard was equilibrated in the LCU for one hour prior to the start of calibration. The zero-air flow was held constant at 1000 sccm, while the gas standard flow was changed every 15 min starting from 40 sccm until 0 sccm in 10 sccm steps. To calibrate at the same humidity level observed in the B2 TRF, 20 μL/min of milli-Q water were dynamically nebulized into the evaporation chamber of the LCU.

Concentrations of compounds not included in the two calibration standard cylinders were calculated applying the kinetic theory of proton transfer reaction with an uncertainty of ≤50%[62].

Only compounds that showed discernable soil fluxes were considered for further analysis. These compounds are reported in Table S2, along with tentative identifications for the underlying VOC species based on previous literature[63]. For methyl nitrite ($CH_3ONO$), the contribution of ion signal from the $^{13}C$ isotopologue of acetic acid ($C_2H_4O_2^+$) at m/z 61.0284 was subtracted from the ion signal at m/z 62.029. The interference of the $^{13}C$ isotopologue also explains the lower mass accuracy for methyl nitrite detection.

In addition, chemical speciation of monoterpene soil fluxes was performed by means of gas chromatography time of flight mass spectrometry (GC-ToF-MS) and three manual soil chambers. Details of the method used for monoterpenes speciation are reported in the supplementary information.

## Soil fluxes measurements

The 12 chambers were placed on PVC-collars (Ø: 20 cm) installed at 2–3 cm depth at four different sites of the B2 TRF eight weeks before the start of the measurements. The sites were on average 25 m away from each other (minimum distance about 10 m and maximum distance about 36 m) and the chambers within each site were placed on average 2.5 m apart (minimum distance about 1.5 m and maximum distance about 4.5 m). When the soil collars were installed, vegetation and litter inside the collars were removed in order to prevent any impact on VOC fluxes from the forest soil. Subsequently, any leaves falling into the chambers were immediately removed. In order to assess background VOC fluxes from the chamber system materials, before the start of the campaign, VOC fluxes were measured from a clean chamber placed on a PFA foil and exposed to the B2TRF ambient air. Three replicates were measured from the blank chamber and mean blank flux and standard deviation for all investigated VOCs are reported in Table S3. Each chamber measurement consisted of 2.5 min of pre-purge during which the chamber lid was open and lines flushed with the ambient air, 6.5 min of closure time and 1 minute of post-purge for a total measurement time of 10 min. All 12 chambers were measured consecutively resulting in a temporal resolution of 2 h. Soil fluxes were calculated from the change in gas concentration in the chamber headspace over the 6.5 min of chamber closure.

VOC fluxes were calculated from the slope obtained by applying the linear regression of the VOC concentration versus the time. Due to the low and bidirectional (emission and uptake) fluxes for most of VOCs, the VOC fluxes were only calculated by applying the linear regression model, similar to the approach used for $N_2O$ fluxes calculations[64]. The first 30 s after chamber closure were discarded due

to possible perturbations induced by the closure and the linear regression was applied to the successive 100 s.

Soil $CO_2$ fluxes were calculated with linear and exponential models, fitted to each individual chamber measurement[64]. In the same way as the VOC fluxes, the first 30 s after chamber closure were omitted and the linear model was applied to the successive 120 s, while the exponential model was applied to the full closure time. The linear model was only used in case the algorithm failed to fit the exponential model.

All the fluxes were corrected for the sampling volume replacement with 100 sccm of synthetic air during the measurements. The fluxes were calculated as:

$$F\left[\mu mol\, m^{-2} s^{-1}\right] = \frac{V\, P}{R\, S\, T}\left(\frac{\Delta C}{\Delta t} + C_0\frac{f}{V}\right) \quad (1)$$

Where $V$ is the chamber volume ($m^3$), $R$ is the gas constant (8.314 $m^3$ Pa $K^{-1}$ $mol^{-1}$), $P$ is the pressure inside the chamber (Pa), T is the chamber air temperature (K), $S$ is the soil surface area within the chamber ($m^2$), $C_0$ is the compound concentration before chamber closure (ppm), $f$ is the synthetic air flow in ($m^3\, s^{-1}$), and $\frac{\Delta C}{\Delta t}$ is the compound concentration change over the time (ppm $s^{-1}$). The term $C0\frac{f}{V}$ in the Eq. (1) represents the correction factor for the sampling volume replacement with synthetic air.

For the linear regression model, the compound concentration (ppm) is plotted against the time (s) and fit with the equation:

$$C(t) = mt + b \quad (2)$$

For the exponential model, only applied to $CO_2$, the data are fitted with the equation:

$$C(t) = C_x + (C_0 + C_x)e^{-a(t-t_0)} \quad (3)$$

Where $C_x$ is a parameter that defines the asymptote and a is $a$ parameter that defines the curvature of the fit. For VOC fluxes, a time factor was applied to convert the results to hourly units.

### $^{13}$C pyruvate labeling

To identify the origin of emitted VOCs, soil was labeled with position specific $^{13}C_1$-pyruvate and $^{13}C_2$-pyruvate. Pyruvate is a central metabolite with high turnover that appears in soils naturally and serves as substrate for primary and secondary metabolic pathways as the $C_1$-carbon position of pyruvate is decarboxylated while the remaining acetyl-CoA can be involved in VOC biosynthesis. The high potential of using position-specific $^{13}$C-labeled pyruvate isotopologues as metabolic tracers to determine qualitative aspects of carbon flux patterns through metabolic pathways has already been demonstrated and exploited either for soil microbial communities[65] and for plants[66,67]. $^{13}$C-pyruvate was added in 9 additional soil chambers located at site 1, site 2, and site 3 (three for each site) of B2 TRF, adjacent to soil chambers measured over the whole campaign. $^{13}$C-pyruvate experiments were performed during pre-drought from 11th to 23th September and during severe drought from 6th to 18th November. Each morning, the chambers were labeled at around 10 AM with $^{13}C_1$-pyruvate or $^{13}C_2$-pyruvate. A 5×5 cm metal frame with 1×1 cm openings was placed into the PVC-collar of each chamber into which 100 µl at concentration 40 mg/ml of $C_1$-$^{13}$C-pyruvate or $C_2$-$^{13}$C-pyruvate solution was added to each 1×1 cm opening to a depth of 1 cm, for a total of 25 injections per chamber[51]. After pyruvate injections, chambers were measured every 30 min for the first 8 h and then were measured every 50 min until 48 h post labeling. The isotopic composition of the flux rate was calculated by applying the linear model to the fractional abundance of $^{13}$C defined as:

$$Fractional\, abundance^{13}C - VOC = \frac{^{13}C - VOC}{(^{13}C - VOC + ^{12}C - VOC)} \quad (4)$$

### Soil properties

Soil moisture, soil temperature and soil matric potential were measured every 15 min by means of two sensors (SMT100, Truebner Gmbh, Neustadt, Germany; TEROS 21, Meter Group, Pullman, WA, USA) installed in 5 cm soil depth from the surface close to the soil chamber sites.

Soil texture was determined using the sedimentation method[68]. Soil bulk density was determined from oven-dry (110 °C) weight of the undisturbed cores of known volume (3 cm in length, 5.7 cm in diameter, Soil moisture Equipment Corp., Goleta, California) and porosity was calculated from bulk density. Total carbon and nitrogen were determined by combustion using Shimadzu TOC-VCSH analyzer (with solid state module SSM-5000A, Columbia, MD). Soil pH and electrical conductivity was determined in 1:1 soil-water suspension with VWR sympHony pH meter (Radnor, PA). For water holding capacity (WHC), 20 g of sieved soil were placed on a glass funnel with a Whatman 40 filter paper. 40 mL of water was added to the plugged funnel to saturate the soil, and after 2 h, the soils were allowed to drain for 6 h. The wet soil was placed on an oven at 105 °C for 48 h to obtain the dry weight of the soil. The WHC was calculated as:

$$WHC\,[\%] = \frac{(Weight\, wet\, soil - Weight\, dry\, soil)}{Weight\, dry\, soil}x\,100 \quad (5)$$

### Statistical analysis

Threshold in the relationship between soil moisture and average of normalized VOC fluxes was identified performing segmented regression analysis. Threshold model was only considered if the Akaike Information Criterion (AIC) of the model was lower than of the corresponding linear model. Threshold model estimation was repeated with 1000 bootstrapped samples to estimate the distribution of the thresholds. The analysis was performed using R (version 4.1.1.) and the package chngpt[69].

Statistical differences between soil fluxes from each site were obtained from Tukey mean comparison test which corrects for Family-wise error rate. Partial least square regression analysis (PLSR) was conducted to assess for covariance between the soil properties and soil VOC fluxes over the rain rewet. PLSR is a multivariate technique used to predict a Y variable (VOC fluxes) with a number of X variables (soil physicochemical properties), which can be correlated. To give equal importance for all variables, the data were centered and all variables were auto-scaled to unit variance. The one-component PLS fitting model was cross-validated using the leave-one-out method. Variables with Variable Importance (VIP) > 1 are considered important while variables with VIP < 1 are considered less important. Tukey mean comparison test and PLSR analysis were performed using OriginPro (Version 2021b, OriginLab Corporation, Northampton, MA, USA).

### Reporting summary

Further information on research design is available in the Nature Portfolio Reporting Summary linked to this article.

## Data availability

All data that support the findings of this study are publicly available in FigShare (https://doi.org/10.6084/m9.figshare.22770782)[70]

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

## Acknowledgements

This research was funded by the European Research Council (ERC; Grant Number 647008 (VOCO2)) to C.W. and from the Philecology Foundation to Biosphere 2 to L.K.M. Gi.Pu. was supported by the German Federal Ministry of Education and Research (BMBF contract 01LB1001A – ATTO +) and the Max Planck Society. L.K.M. was supported by the National Science Foundation under Grant No. 2045332. We like to thank Dr Linnea Honeker for her help on collecting and processing soil samples.

## Author contributions

Gi.Pu. analyzed the data, prepared and interpreted the results, and wrote the paper. J.I., L.K.M., T.K., Jv.H., J.K., C.W., J.W. designed the experimental set-up. J.I., L.K.M., E.Y.P., K.M., J.B., Ge.Pu., J.G.L., K.D., S.N.L., C.W. performed the experiments. S.N.L., L.K.M., C.W. and J.W. supervised and conceived the experiments. All authors contributed to writing and editing the manuscript.

## Funding

## Competing interests

The authors declare no competing interests
