## [Peer Review File · Nature Communications]

Effects of drought and recovery on soil volatile organic compound fluxes in an experimental rainforestReviewer #1 (Remarks to the Author):

Pugliese et al. report on VOC fluxes from soils in an experimental rain forest. The BIOSPHERE-2 experimental forest is a unique facility that enables measurements under controlled conditions to unravel the drivers of VOC fluxes from rain forest vegetation and soils. The use of ^{13}C labeling in situ provides new insights on the microbial role in VOC fluxes. The set of VOC measurements reported in this manuscript is novel and interesting to understand the reactive carbon dynamics of rain forests during drought events. One of the main questions I have is about the apparent absence of blank soil chambers used for the soil flux measurements. In addition, I provide some comments to improve the manuscript.

MAJOR COMMENTS

Lines 124 and 131. The authors talk about "increased ambient air concentrations". What do they think was the cause of higher concentrations in the air? Do the plant emissions measured concurrently with soil fluxes indicate that plant emissions caused the increased air concentrations? Since on some days there were net emissions and, on others, net uptake, can the authors calculate a "compensation point" (the ambient concentration below which release occurs, and above which uptake occurs)?

Line 238. "one order of magnitude higher". Do this refers to the relative abundance of ^{13}C -enriched acetone? Or to the absolute magnitude of ^{13}C -enriched emissions? I think it would be interesting to know not only how the relative (i.e, $^{13}\text{C}/[^{13}\text{C}+^{12}\text{C}]$) flux changes, but also the absolute flux (e.g., in micromol/m²/h).

Lines 322-324 + 332-337. The soil chamber system is described as a "dynamic" system. This normally implies the continuous introduction of "fresh" air to the system, so that a somewhat steady-state of VOC concentration is achieved inside the chamber. However, after reading the description, it seems that during the measurements the air is recirculated inside the chamber system, making it effectively a closed system. I suggest that the authors clearly explain how the chamber system worked, as not everyone is familiar with those particular models of chambers. Also, list which modifications, if any, were performed onto the commercial soil chambers to allow the sampling by the PTR-TOF-MS.

Lines 374-378. Related to the previous point, do the authors have a bibliographic reference to document the "linear regression model" that they used? And, furthermore, was any correction introduced in the calculations to account for the 100 sccm of synthetic air added during the measurements? The same applies to CO₂ measurements (line 379, "linear and exponential models").

Related to the chamber system, did the authors use a blank chamber (same chamber materials as the other chambers, but without soil inside)? This is not mentioned in the manuscript but it is typically essential to have such a blank, to be able to exclude any apparent VOC fluxes that may be a result of the chamber materials (adsorption or desorption of VOCs, for example).

MINOR COMMENTS

Line 39. The introduction indicates that the contribution of soil VOC to total ecosystem budget can even be "comparable to that of the plants". The authors participated in measurements not only of soils, but probably of plants too during this study at Biosphere-2. Were the VOC fluxes from the soils comparable to those of plants?

Line 71. What does exactly this text mean: "with little diel dynamics (1.4 +- 4 degC)"? Please clarify what this means.

Lines 146-156. This paragraph can be hard to follow, I suggest improving its readability.

Line 293. Should the content of clay range between 20-30%? If it can be up to 35%, then the sand content cannot be >70% ($35+70 > 100\%$).

Line 360. Should reference number 66 be Yanez-Serrano et al 2021 (doi 10.1016/j.atmosenv.2020.117929) instead of Yanez-Serrano et al 2020?

Acknowledgements and author contributions. "G.P." (or GP) can be ambiguously attributed to both authors Giovanni Pugliese and Gemma Purser. Please disambiguate the initials used in these sections.

Figure 1. The background colors are hard to distinguish. I suggest, at a minimum, to include the DOY of the phase changes in the caption.

Figure 2. I suggest providing (e.g., in the supplement) a figure with the diel cycles of fluxes for each phase of drought, like Figure 5 but for each phase. Also, for VOC concentrations in ambient air outside the chambers.

Figure 4. Are the traces shown in this graph an average of several chambers? This is not mentioned in the caption. If they are indeed averages, some sort of indication of dispersion of data (e.g., standard deviation) would be useful, although it may be difficult to include too much data in the figure.

Figure 7. Upper panel and lower panel content (drought vs pre-drought) is not consistent between the plot and the caption. Please correct.

Supporting information. The last sentence needs editing ("Samples were collected at The inert coated...").

Reviewer #2 (Remarks to the Author):

In this article, the authors describe the effect of prolonged drought and rewetting on soil VOC fluxes in controlled conditions. The authors describe that under wet conditions, rainforest soil acts as a net sink of VOCs, in particular for isoprenoids, carbonyls and alcohols. However, this capacity decreases with an increase in drought conditions and at a certain level, the soil becomes the source of VOCs, which was related to soil microbial activity. The overall outcome of the study is not novel in a way that similar studies have been reported earlier (i.e., Monard et al. 2021, Gray et al. 2014, Trowbridge et al. 2020, Raza et al. 2017); however, this study is relatively more comprehensive and describes dynamics of some VOCs fate in soil under drought and rewetting condition. There are several main factors that were overlooked while performing this study which reduces its comprehensiveness.

It's all about soil uptake and emission but no analysis was conducted for soil VOCs, and some conclusions were speculated based on the patterns of the emission of VOCs without really describing whether it is the microbial activity or soil physicochemical characteristics playing a role.

Microbes are stated as the main contributor to soil sink activity but no true evidence is reported. In addition, inducing soil microbial activity by adding a foreign substrate does not really reflect the soil process under natural conditions but can only aid in conclusions.

Some VOCs showed increased emission just after rain events indicating the role of soil properties in it. Soil physicochemical properties, hygroscopicity of soil, pore size, clay particles, organic matter, water holding capacity? These all have an important role in the retention of VOCs in soil, especially organic matter adsorbs more VOCs under wet conditions while clay particles adsorb more VOCs under dry conditions (Ong and Lion, 1991). What about the uptake of VOCs by plants?

Does air velocity under natural conditions have some role to play?

Role of temperature is overlooked, which is not only directly related to climate change but also to microbial activity and retention properties of VOCs.

Microbial activity was thought to be responsible for pulses of some VOCs like Dimethyl disulfide, is it possible for microbes to produce pulses of any VOC just after rewetting in a short time or there are some other abiotic factors involved?

**Any relationship of outcomes with VOCs mass or class or retention properties?
How do leaves or debris cover in tropical forests contribute to soil VOC flux? This is the ignored portion of this study.**

Number of VOCs identified in soil seems low in numbers.

VOC production is highly sensitive to the nutrient's sources, does the addition of ¹³C-labelled pyruvate hold the merit of the natural process of soil?

Minor issues

Starting title with 'The' is not appropriate and also the use of the abbreviation 'VOC'.

Line 36-38: That is an outdated statement

Line 43-45: Is there any role soil minerals or particles (i.e., clay) play in this process?

Line 52-53: What about temperature rise? It is an important component of climate change.

Line 55: rainforest contribution 70%? Soil or plant or both?

Line 110: Abiotic dissolution is underestimated term for VOCs fate in the soil overlooking soil physical properties' role in the absorption or adsorption of VOCs in soil.

Line 161: "19% represents the soil moisture". This conclusion is not very important without exploring soil properties contribution.

Line 224-226: What could be the reason for the depletion of isoprenoids in ambient air at night?

line 273: sudden use of an abbreviation

Line 308: rewet by adding ~2.2 L (~22.5 mm) of water per chamber? does the sudden application of water affect osmotic shock or VOC emission properties?

Line 345: How many VOCs are in that mixture?

Line 296: The enclosed air is therefore relatively rich in primary VOC emissions and relatively poor in oxidized products. So, do natural conditions already compromised?

Reviewer #3 (Remarks to the Author):

The manuscript by Pugliese and co-authors explore soil VOCs fluxes during drought and during post drought recovery in an experimental rainforest. The manuscript is well in line with the quite recent interest from the community on soil VOCs emissions. The manuscript is well written, and the results are well presented. In particular, the authors observed emissions pulse of dimethyl sulfide after soil rewatering, and emissions of methyl nitrite under very severe drought conditions. These are interesting results as these two compounds are usually associated with oceanic emissions and are highly reactive in the atmosphere. To my knowledge, this is the first time that soil methyl nitrite emissions are observed.

However, the authors failed to convince me that their results imply a significant impact of soil VOC on atmospheric chemistry and climate and the relative gain of adding soil VOC in land surface models:

lines 29-30 'Results show that, the extended drought periods predicted for tropical rainforest regions will strongly affect soil VOC fluxes thereby impacting atmospheric chemistry and climate'.

Lines 284-288 'Prolonged drought and recovery had a major impact on soil VOC fluxes from the experimental rainforest, affecting the composition and quantity of VOCs in the atmosphere of the enclosed ecosystem. Soil VOC fluxes and their parametrization related to soil moisture levels must be included in atmospheric models to simulate current atmospheric chemistry and to improve climate model predictions of ecosystem responses to drought'.

Indeed, the authors found a pulse of Dimethyl sulfide about 0.2 $\mu\text{mol}/\text{m}^2/\text{h}$ lasting for less than 10 days after rewatering. It seems relatively small when compared to oceanic fluxes which annual mean varies roughly between 0.15 and 0.35 $\mu\text{mol}/\text{m}^2/\text{h}$ and last all year around (cf. Wang, S., Maltrud, M., Elliott, S. et al. Influence of dimethyl sulfide on the carbon cycle and biological production. *Biogeochemistry* 138, 49–68 (2018). <https://doi.org/10.1007/s10533-018-0430-5>).

The same applies for methyl nitrite. Maximum of soil emissions showed in the manuscript corresponds to annual mean flux in equatorial oceans (Fisher, J. A., Atlas, E. L., Barletta, B., Meinardi, S., Blake, D. R., Thompson, C., et al. (2018). Methyl, ethyl, and propyl nitrates: Global distribution and impacts on reactive nitrogen in remote marine environments. *Journal of Geophysical Research: Atmospheres*, 123, 12,429– 12,451. <https://doi.org/10.1029/2018JD029046>). However, this maximum of emissions are found under extreme drought conditions, corresponding to a reduction of more than 50% of soil moisture (Figure 1). This reduction of soil moisture must be put in context. As the authors stated line 256-258, soil moisture anomaly were of almost 30% during the strong El Niño drought in 2015/2016. We are therefore still far away from a reduction of 50% of soil moisture, which seems to me very close to the permanent wilting point.

To address the importance of their findings in term of impact, I wish the authors would have compared their findings with what we know from oceanic fluxes studies. That might have moderated (or not) their conclusion of the necessity to incorporate soil VOC fluxes into land surface models.

Consequently, it is difficult for me to assess if the manuscript is relevant for Nature Communications.

Dear Reviewers,

Thank you very much for your time and consideration on our manuscript. Your comments were extremely helpful and by addressing all of them the manuscript has substantially improved.

In the blue text below, we give first a general discussion of the main points raised and then point-by-point discussions of all the questions of the reviewers noted in bold and black. Unmodified text of the manuscript is reported in plain black and all resultant changes to the manuscript are marked in red.

Sincerely,

Dr. Giovanni Pugliese

Corresponding author

REVIEWER COMMENTS

Reviewer #1 (Remarks to the Author):

Pugliese et al. report on VOC fluxes from soils in an experimental rain forest. The BIOSPHERE-2 experimental forest is a unique facility that enables measurements under controlled conditions to unravel the drivers of VOC fluxes from rain forest vegetation and soils. The use of ¹³C labeling in situ provides new insights on the microbial role in VOC fluxes. The set of VOC measurements reported in this manuscript is novel and interesting to understand the reactive carbon dynamics of rain forests during drought events. One of the main questions I have is about the apparent absence of blank soil chambers used for the soil flux measurements. In addition, I provide some comments to improve the manuscript.

Thank you for this positive feedback, and for highlighting that this study provides new insights on the microbial role in VOC fluxes and reactive carbon dynamics of rain forests during drought events. All points raised are addressed in detail below.

MAJOR COMMENTS

Lines 124 and 131. The authors talk about "increased ambient air concentrations". What do they think was the cause of higher concentrations in the air? Do the plant emissions measured concurrently with soil fluxes indicate that plant emissions caused the increased air concentrations?

At line 124 we were referring to the increased ambient air concentrations observed for the two alcohols, methanol and ethanol, during the recovery period. Methanol and ethanol are synthesized in plants and emitted in the atmosphere from leaves and stems. Increased methanol and ethanol emission from plants have been observed previously under stress conditions such as drought, flooding or leaf damage (Kirstine & Galbally, 2012; Dorokov *et al.*, 2018). Therefore, the increased concentration of the alcohols in the B2TRF ambient air during the recovery period was likely induced by the rewetting of plant leaves and litter. In contrast, at line 131 we were referring to dimethyl sulfide concentrations in ambient air that peaked after the rain rewet on the days (day 346 and 347) when the highest dimethyl sulfide emissions from the soil were observed. Therefore, for dimethyl sulfide we concluded that the soil significantly contributed to the dimethyl sulfide concentrations in the B2TRF ambient air after the rain rewet.

VOC emission fluxes (i.e. isoprene, monoterpenes) from specific plants were measured from branch cuvettes during the B2WALD campaign, however the data are not yet fully available and they will be the topic of a future publication. Data on isoprene and monoterpene emission indicate that plants considerably contribute to the atmospheric concentrations in the B2TRF. Periods with elevated atmospheric abundances of these gases correlate with periods of high emission rates of one of the most important species in the ecosystem (*Clitoria* trees).

References:

Kirstine, W. V. & Galbally, I. E. The global atmospheric budget of ethanol revisited. *Atmospheric Chemistry and Physics* 12, 545–555 (2012).

Dorokhov, Y. L., Sheshukova, E. V. & Komarova, T. V. Methanol in Plant Life. *Frontiers in Plant Science* 9, (2018).

We integrate the reviewer's point by including the following text in the manuscript.

Changes in the manuscript:

3.1 Long term soil VOC fluxes dynamics

The higher soil uptake capacity observed for both alcohols during the recovery period could be due to the synergistic effect of increased microbial activity, increased abiotic dissolution in wet soil and increased ambient air concentrations (Figure S1) **attributable to higher plant leaf and litter emissions induced by rewet (Kirstine & Galbally, 2012; Dorokov *et al.*, 2018).**

Since on some days there were net emissions and, on others, net uptake, can the authors calculate a "compensation point" (the ambient concentration below which release occurs, and above which uptake occurs)?

Thanks for this interesting comment. The compounds that were both emitted and taken-up by the soil were C₅H₈O, acetone, acetaldehyde, butanone and pentanone. We tested for a relationship between soil fluxes and ambient concentrations to determine a compensation point for these compounds, but we did not find a significant relationship. Soil fluxes of these compounds were mainly driven by soil moisture levels, which dramatically changed over the campaign, masking the influence of the ambient concentrations.

We consider the reviewer's point by including the following text in the manuscript:

3.1 Long term soil VOC fluxes dynamics

It should be noted that **soil VOC fluxes** are both a function of soil processes and ambient concentrations above the soil (Figure S1). **For VOCs that were both emitted and taken up by the soil, namely C₅H₈O, acetone, acetaldehyde, butanone, and pentanone, no relationship was found between soil fluxes and respective ambient concentrations. Thus, no fixed compensation point, i.e., the ambient VOC concentration at which the soil flux is zero, could be identified. This is because the soil fluxes for these VOCs was mainly driven by soil moisture levels, which dramatically changed over the campaign, masking the influence of the ambient concentrations..**

Line 238. "one order of magnitude higher". Do this refers to the relative abundance of ¹³C-enriched acetone? Or to the absolute magnitude of ¹³C-enriched emissions? I think it would be interesting to know not only how the relative (i.e, ¹³C/[¹³C+¹²C]) flux changes, but also the absolute flux (e.g., in micromol/m²/h).

It refers to the fractional abundance of ¹³C-acetone but we agree that the statement can cause confusion to the reader. Therefore, we clarified this by changing the sentence and moreover, as the reviewer suggested, we added a new figure in the Supplementary Information (Figure S6) to also show the absolute flux of ¹³C-acetone, both during pre-drought and drought.

Changes in the manuscript:

3.4 Origin of VOC emissions

To identify the origin of the emitted VOCs, the soil was labeled with position specific $^{13}\text{C}_1$ -pyruvate and $^{13}\text{C}_2$ -pyruvate. A net soil emission was observed for the flux of the fractional abundance of ^{13}C -acetone (defined as $^{13}\text{C-VOC}/(^{13}\text{C-VOC} + ^{12}\text{C-VOC})$), after $^{13}\text{C}_2$ -pyruvate injections both during pre-drought and during drought period (Figure 7). Absolute fluxes of ^{13}C -acetone are shown in Figure S6. This is clear evidence that soil microbes are able to produce VOCs from precursors in the soil and that the emissions observed were not just due to abiotic release from soil. As shown in Figure 2, acetone was mainly consumed under wet soil conditions; therefore, the emission observed for the fractional abundance of ^{13}C -acetone during pre-drought demonstrated that soil microbes can both produce and consume acetone and that under wet conditions they were able to consume more acetone than they actually produced. During drought the emission fluxes of the fractional abundance of ^{13}C -acetone was about one order of magnitude higher than during pre-drought. This is further evidence that under drought stress soil microbes used energy resources to support higher VOC production.

Figure 7 Soil emission fluxes of the fractional abundance of ^{13}C -acetone (defined as $^{13}\text{C}\text{-VOC}/(^{13}\text{C}\text{-VOC} + ^{12}\text{C}\text{-VOC})$) after the $\text{C}_1\text{-}^{13}\text{C}$ -pyruvate (blue lines) and $\text{C}_2\text{-}^{13}\text{C}$ -pyruvate (red lines) soil injections during pre-drought (upper panel) and drought (lower panel) period. Lines represent averaged fluxes over the 9 chambers. The shaded areas indicate the standard deviation.

Figure S6 Soil emission fluxes of ^{13}C -acetone after the $\text{C}_1\text{-}^{13}\text{C}$ -pyruvate (blue lines) and $\text{C}_2\text{-}^{13}\text{C}$ -pyruvate (red lines) soil injections during pre-drought (upper panel) and drought (lower panel) period. Lines represent averaged fluxes over the 9 chambers. The shaded areas indicate the standard deviation.

Lines 322-324 + 332-337. The soil chamber system is described as a "dynamic" system. This normally implies the continuous introduction of "fresh" air to the system, so that a somewhat steady-state of VOC concentration is achieved inside the chamber. However, after reading the description, it seems that during the measurements the air is recirculated inside the chamber system, making it effectively a closed system. I suggest that the authors clearly explain how the chamber system worked, as not everyone is familiar with those particular models of chambers.

Also, list which modifications, if any, were performed onto the commercial soil chambers to allow the sampling by the PTR-TOF-MS.

The reviewer is correct, the chamber system used was a closed system. The ambient air was sampled during the pre-purge period (2.5 minutes) with chamber lid open, and then once the chamber lid was closed, the enclosed air was recirculated inside the system during the flux measurement period (6.5 minutes). The chamber system was adapted to allow sampling with different analyzers, including the PTR-TOF-MS for VOC flux measurements, by introducing a T-piece on the outflow of the LI-8100A. 100 sccm were sampled from the outflow and distributed to the different analyzers. To avoid negative pressure, ca. 100 sccm of VOC-free synthetic air was introduced in the soil flux system at the inflow of the LI-8100A. Moreover, in order to minimize surface effects on VOC analysis, perfluoroalkoxy (PFA) tubing was used for the soil flux system, for the subsampling line and for the PTR inlet. We consider the reviewer's point by including the following text:

5.2 Experimental set-up

Soil VOC fluxes were measured continuously using a proton transfer time of flight mass spectrometer (PTR-ToF-MS-8000, Ionicon Analytik GmbH, Innsbruck, Austria) directly connected to the outflow of an automated soil flux system consisting of a LI-8100 infrared gas analyzer (IRGA; for CO₂ fluxes measurement), a LI-8150 16-port multiplexer (Licor Inc., Lincoln, NA, USA) and 12 dynamic soil flux chambers (LI 8100-104 Long-Term Chambers with opaque lids, Licor Inc.). **The soil flux measurement is based on a closed dynamic system. The ambient air was first sampled during the pre-purge period (2.5 minutes) with chamber lid open, and then once the chamber lid was closed, the enclosed air was recirculated inside the system during the flux measurement period (6.5 minutes). The soil flux measurement is based on the calculation of the compound concentration development in the recirculated ambient air. The soil chambers are featured with a pressure vent system to maintain pressure equilibrium inside the chambers even under windy conditions.**

A detailed description and the working principle of the PTR-ToF-MS instrument can be found elsewhere⁶². Concisely, the soft ionization process is based on a proton transfer from hydronium ions (H₃O⁺) to sample VOCs having a higher proton affinity than water (691 kJ mol⁻¹). Protonated VOCs are then analyzed in a high-resolution time-of-flight mass spectrometer according to mass-to-charge ratio (m/z). The instrumental settings were as follows: the PTR drift tube pressure was 2.2 mbar, the PTR drift tube voltage was 600 V, and the PTR drift tube temperature was 60 °C, resulting in an E/N ratio of 137 Td. The time resolution was 10 s with the m/z monitored up to 500 Da. The total volume of the soil flux system, including chamber, tubing, IRGA and multiplexer, was about 6.5-7 L. For gas analysis, ca. 100 sccm were subsampled **via a T-piece placed at outflow** the LI-8100A and **were** distributed to the different analyzers including the PTR for VOCs measurement. To avoid negative pressure, ca. 100 sccm of synthetic air were introduced in the soil flux system **via a T-piece placed at the inflow of the Li-8100A**. In order to minimize surface effects on VOC analysis, perfluoroalkoxy (PFA) tubing was used for the soil flux system, for the subsampling line and for the PTR inlet.

Lines 374-378. Related to the previous point, do the authors have a bibliographic reference to document the "linear regression model" that they used? And, furthermore, was any correction introduced in the calculations to account for the 100 sccm of synthetic air added during the measurements? The same applies to CO₂ measurements (line 379, "linear and exponential models").

In studies where non-steady state chambers are used, CO₂ efflux is usually estimated from the rate of change of CO₂ concentrations in relation to time using linear or exponential models (e.g., Barba *et al.*, 2019). For the VOC flux calculations we applied the linear regression model due to the low and bidirectional (emission and uptake) fluxes for most of VOCs similar to the approach used for N₂O fluxes calculations in Barba *et al.* 2019. The fluxes were corrected for the sampling volume replacement with 100 sccm of synthetic air during the measurements. Therefore, the fluxes were calculated as:

$$F [\mu\text{mol m}^{-2} \text{s}^{-1}] = \frac{V P}{R S T} \left(\frac{\Delta C}{\Delta t} + C_0 \frac{f}{V} \right) \quad (1)$$

Where V is the chamber volume (m³), R is the gas constant (8.314 m³ Pa K⁻¹ mol⁻¹), P is the pressure inside the chamber (Pa), T is the chamber air temperature (K), S is the chamber surface area in (m²), C_0 is the compound concentration before chamber closure (ppm), f is the synthetic air flow in m³ s⁻¹, and $\frac{\Delta C}{\Delta t}$ is the compound concentration change over the time (ppm s⁻¹). The term $C_0 \frac{f}{V}$ in the equation (1) represents the correction factor for the sampling volume replacement with synthetic air.

For the linear regression model, the compound concentration (ppm) is plotted against the time (s) and fit with the equation:

$$C(t) = mt + b \quad (2)$$

For the exponential model, the data are fitted with the equation:

$$C(t) = C_x + (C_0 + C_x)e^{-a(t-t_0)} \quad (3)$$

Where C_x is a parameter that defines the asymptote and a is a parameter that defines the curvature of the fit. For VOC fluxes, a time factor was applied to convert the results to hourly units.

Reference:

Barba, J., Poyatos, R. & Vargas, R. Automated measurements of greenhouse gases fluxes from tree stems and soils: magnitudes, patterns and drivers. *Sci Rep* 9, 4005 (2019).

We bring the reviewer's point by including the following text in the manuscript:

5.3 Soil fluxes measurements

.....All 12 chambers were measured consecutively resulting in a temporal resolution of 2 hours. Soil fluxes were calculated from the change in gas concentration in the chamber headspace over the 6.5 minutes of chamber closure.

VOC fluxes were calculated from the slope obtained by applying the linear regression of the VOC concentration versus the time. Due to the low and bidirectional (emission and uptake) fluxes for most of VOCs, the VOC fluxes were only calculated by applying the linear regression model, similar to the approach used for N₂O fluxes calculations (Barba *et al.* 2019). The first 30 s after chamber closure were discarded due to possible perturbations induced by the closure and the linear regression was applied to the successive 100 s.

Soil CO₂ fluxes were calculated with linear and exponential models, fitted to each individual chamber measurement (Barba *et al.*, 2019). In the same way as the VOC fluxes, the first 30 s after chamber closure were omitted and the linear model was applied to the successive 120 s, while the exponential model was applied to the full closure time. The linear model was only used in case the algorithm failed to fit the exponential model.....

All the fluxes were corrected for the sampling volume replacement with 100 sccm of synthetic air during the measurements. The fluxes were calculated as:

$$F [\mu\text{mol m}^{-2} \text{s}^{-1}] = \frac{VP}{RST} \left(\frac{\Delta C}{\Delta t} + C_0 \frac{f}{V} \right) \quad (1)$$

Where V is the chamber volume (m^3), R is the gas constant ($8.314 \text{ m}^3 \text{ Pa K}^{-1} \text{ mol}^{-1}$), P is the pressure inside the chamber (Pa), T is the chamber air temperature (K), S is the chamber surface area in (m^2), C_0 is the compound concentration before chamber closure (ppm), f is the synthetic air flow in $\text{m}^3 \text{ s}^{-1}$, and $\frac{\Delta C}{\Delta t}$ is the compound concentration change over the time (ppm s^{-1}). The term $C_0 \frac{f}{V}$ in the equation (1) represents the correction factor for the sampling volume replacement with synthetic air.

For the linear regression model, the compound concentration (ppm) is plotted against the time (s) and fit with the equation:

$$C(t) = mt + b \quad (2)$$

For the exponential model, only applied to CO_2 , the data are fitted with the equation:

$$C(t) = C_x + (C_0 + C_x)e^{-a(t-t_0)} \quad (3)$$

Where C_x is a parameter that defines the asymptote and a is a parameter that defines the curvature of the fit. For VOC fluxes, a time factor was applied to convert the results to hourly units.

Related to the chamber system, did the authors use a blank chamber (same chamber materials as the other chambers, but without soil inside)? This is not mentioned in the manuscript but it is typically essential to have such a blank, to be able to exclude any apparent VOC fluxes that may be a result of the chamber materials (adsorption or desorption of VOCs, for example).

We agree with the reviewer on the importance of the blank chamber fluxes in order to assess background VOC fluxes from the soil flux system materials. Before the start of the campaign, we measured blank fluxes from a clean chamber placed on a PFA foil and exposed to the B2TRF ambient air. We measured three replicates and mean blank flux and standard deviation for all investigated VOCs are shown in table 1. Results indicate that blank fluxes were negligible compared to the soil fluxes for all investigated VOCs (see figure2).

Table 1 Blank fluxes for all investigated VOCs. Mean flux and SD represent the average and the standard deviation, respectively, over three replicate measurements.

VOC	Mean flux (n=3) [$\mu\text{mol m}^{-2} \text{h}^{-1}$]	SD [$\mu\text{mol m}^{-2} \text{h}^{-1}$]
Isoprene	-0.20211	0.24616
$\text{C}_5\text{H}_8\text{O}$	0.00928	0.00442
Monoterpenes	0.00880	0.01153
Methacrolein	0.00272	0.00090
Acetone fluxes	0.05415	0.00484
Acetaldehyde	0.01805	0.00394
Butanone	0.00923	0.00714

Pentanone	0.01138	0.00326
Methanethiol	0.00990	0.00360
Dimethyl sulfide	-0.00100	0.01353
C₄H₁₀S	-0.00009	0.00567
C₃H₈OS	0.02643	0.00581
Methanol	-0.10493	0.03807
Ethanol	0.11547	0.10256
Methyl nitrite	0.00095	0.00422

In light of the comment we have added the description of blank fluxes measurement in the method section and we added table 1 in the supplementary information as table S3.

Changes in the manuscript:

5.3 Soil fluxes measurements

.....In order to assess background VOC fluxes from the chamber system materials, before the start of the campaign, VOC fluxes were measured from a clean chamber placed on a PFA foil and exposed to the B2TRF ambient air. Three replicates were measured from the blank chamber and mean blank flux and standard deviation for all investigated VOCs are reported in Table S3. Each chamber measurement consisted of 2.5 minutes of pre-purge during which the chamber lid was open and lines flushed with the ambient air, 6.5 minutes of closure time and 1 minute of post-purge for a total measurement time of 10 minutes. All 12 chambers were measured consecutively resulting in a temporal resolution of 2 hours....

SUPPLEMENTARY INFORMATION

Table S 3 Blank fluxes for all investigated VOCs. Mean flux and SD represent the average and the standard deviation, respectively, over three replicate measurements.

VOC	Mean flux (n=3) [$\mu\text{mol m}^{-2} \text{h}^{-1}$]	SD [$\mu\text{mol m}^{-2} \text{h}^{-1}$]
Isoprene	-0.20211	0.24616
C₅H₈O	0.00928	0.00442
Monoterpenes	0.00880	0.01153
Methacrolein	0.00272	0.00090
Acetone fluxes	0.05415	0.00484
Acetaldehyde	0.01805	0.00394
Butanone	0.00923	0.00714
Pentanone	0.01138	0.00326
Methanethiol	0.00990	0.00360
Dimethyl sulfide	-0.00100	0.01353

C₄H₁₀S	-0.00009	0.00567
C₃H₈OS	0.02643	0.00581
Methanol	-0.10493	0.03807
Ethanol	0.11547	0.10256
Methyl nitrite	0.00095	0.00422

MINOR COMMENTS

Line 39. The introduction indicates that the contribution of soil VOC to total ecosystem budget can even be "comparable to that of the plants". The authors participated in measurements not only of soils, but probably of plants too during this study at Biosphere-2. Were the VOC fluxes from the soils comparable to those of plants?

We were referring to previously published studies (Kramshøj *et al.*, 2016, Bourtsoukidis *et al.*, 2018 and Staudt *et al.*, 2019) which showed that in certain conditions soil emissions were comparable to that of the plants. Regarding the B2WALD campaign, as stated above, plant VOC emissions were measured and we completely agree with the reviewer that it is interesting to compare them with soil VOC fluxes, however, that data will be the topic of another paper.

Line 71. What does exactly this text mean: "with little diel dynamics (1.4 +- 4 degC)"? Please clarify what this means.

It means that the differences between daytime and nighttime soil temperature were small, i.e., on average 1.4 +- 4 °C. We agree that the statement may be not clear to the reader and therefore we modified the sentence.

Changes in the manuscript:

3.1 Long term soil VOC fluxes dynamics

Soil moisture (Figure 1, violet plot) and soil matric potential (Figure 1, orange plot) decreased strongly as the drought progressed, from 29 to 12.5% and from 0 to -3.7 MPa, respectively, but recovered back to pre-drought levels after the rewetting rain events. As expected, the soil temperature was relatively stable throughout the campaign (21.5- 25.5°C) and it showed a low diel variation with nighttime temperatures on average 1.4 ± 0.4 °C lower than the daytime temperatures.

Lines 146-156. This paragraph can be hard to follow, I suggest improving its readability.

We improved the readability of the paragraph as suggested.

Changes in the manuscript:

3.1 Long term soil VOC fluxes dynamics

It should be noted that soil VOC fluxes are both a function of soil processes and ambient concentrations above the soil (Figure S1). For VOCs that were both emitted and taken up by the soil, namely C₅H₈O, acetone, acetaldehyde, butanone, and pentanone, no relationship was found between soil fluxes and respective ambient concentrations. Thus, no fixed compensation point, i.e., the ambient VOC concentration at which the soil flux is zero, could be identified. This is because the soil fluxes for these VOCs was mainly driven by soil moisture levels, which dramatically changed over

the campaign, masking the influence of the ambient concentrations. Nevertheless, to assess the effect of the VOC concentrations in the ambient air on VOC soil uptake rates, we calculated the deposition velocities, which are defined as the ratio of VOC uptake rates to their ambient concentrations. The trends in deposition velocities (Figure S3) were very similar to those of the net uptake fluxes (Figure 2). However, during the recovery period, net soil uptake rates of isoprene were lower compared to pre-drought period (Figure 2), while isoprene soil deposition velocity had actually returned to pre-drought levels (Figure S3). This indicates that the lower net isoprene soil uptake during the recovery period compared to the pre-drought period was due to a lower isoprene concentration in the ambient air (Figure S1) and not to a reduction in microbial uptake capacity.

Line 293. Should the content of clay range between 20-30%? If it can be up to 35%, then the sand content cannot be >70% (35+70 > 100%).

Thanks for your comment. We deleted this sentence and we reported detailed soil physicochemical properties for each of the four sites of the B2TRF.

Changes in the manuscript:

Table 1 Soil physicochemical properties at each site of the B2 TRF where soil flux chambers were placed.

Site	S1	S2	S3	S4
Texture	Loam	Loam	Loam	Silt Loam
% Clay, <2 μm	17.90 \pm 1.74	23.75 \pm 0.00	17.93 \pm 0.03	25.07 \pm 0.19
% Silt, 2-50 μm	46.13 \pm 3.41	36.25 \pm 0.00	44.80 \pm 0.15	51.37 \pm 0.73
% Sand, 50 - 2000 μm	35.99 \pm 4.96	39.24 \pm 2.88	37.27 \pm 0.18	23.57 \pm 0.91
% Gravel	19.79	15.12	26.36	19.06
Bulk Density, g/cm ³	1.38 \pm 0.20	1.40 \pm 0.21	1.21 \pm 0.29	1.40 \pm 0.15
Porosity	0.48 \pm 0.07	0.47 \pm 0.08	0.54 \pm 0.11	0.47 \pm 0.08
Total Carbon, $\mu\text{g}/\text{mg}$	24.10 \pm 8.02	27.00 \pm 3.06	29.40 \pm 7.30	21.44 \pm 6.51
Total Nitrogen, $\mu\text{g}/\text{mg}$	1.90 \pm 0.69	2.26 \pm 0.18	2.54 \pm 0.80	1.70 \pm 0.42
Electrical conductivity, $\mu\text{S}/\text{cm}$	409.10 \pm 130.7	771.30 \pm 102.4	644.00 \pm 277.90	257.10 \pm 77.80
pH	7.26 \pm 0.08	7.44 \pm 0.14	7.31 \pm 0.11	6.98 \pm 0.32
Water holding capacity, %	66.94 \pm 6.70	60.48 \pm 3.38	59.03 \pm 2.82	70.78 \pm 4.85

5.5 Soil properties

Soil moisture, soil temperature and soil matric potential were measured every 15 minutes by means of two sensors (SMT100, Truebner GmbH, Neustadt, Germany; TEROS 21, Meter Group, Pullman, WA, USA) installed in 5 cm soil depth from the surface close to the soil chamber sites.

Soil texture was determined using the sedimentation method. Soil bulk density was determined from oven-dry (110 °C) weight of the undisturbed cores of known volume (3 cm in length, 5.7 cm in diameter, Soil moisture Equipment Corp., Goleta, California) and porosity was calculated from bulk density. Total carbon and nitrogen were determined by combustion using Shimadzu TOC-VCSH analyzer (with solid state module SSM-5000A, Columbia, MD). Soil pH and electrical conductivity was determined in 1:1 soil-water suspension with VWR sympHony pH meter (Radnor, PA). For water holding capacity (WHC), 20 g of sieved soil were placed on a glass funnel with a Whatman 40 filter paper. 40 mL of water was added to the plugged funnel to saturate the soil, and after 2 h, the soils were allowed to drain for 6 h. The wet soil was placed on an oven at 105 °C for 48 h to obtain the dry weight of the soil. The WHC was calculated as:

$$WHC [\%] = \frac{(Weight\ wet\ soil - Weight\ dry\ soil)}{Weight\ dry\ soil} \times 100 \quad (4)$$

Line 360. Should reference number 66 be Yanez-Serrano et al 2021 (doi 10.1016/j.atmosenv.2020.117929) instead of Yanez-Serrano et al 2020?

Yes, the reviewer is right. We changed the reference accordingly.

Acknowledgements and author contributions. "G.P." (or GP) can be ambiguously attributed to both authors Giovanni Pugliese and Gemma Purser. Please disambiguate the initials used in these sections.

Thanks, we now used Gi.Pu for Giovanni Pugliese and Ge.Pu for Gemma Purser.

Figure 1. The background colors are hard to distinguish. I suggest, at a minimum, to include the DOY of the phase changes in the caption.

As reviewer suggested, in Figure 1, Figure 2, Figure S1, Figure S2 and Figure S3 we made background colors darker in order to be more distinguishable and we also added the DoY of phase change in the captions.

Figure 2. I suggest providing (e.g., in the supplement) a figure with the diel cycles of fluxes for each phase of drought, like Figure 5 but for each phase. Also, for VOC concentrations in ambient air outside the chambers.

As the reviewer suggested, we modified Figure 5 adding diel cycles for each phase of drought. We did the same also for Figure S4 that shows the diel cycle of oxygenated compounds for each drought phase. As we added one additional figure to the manuscript and one additional figure to the supplementary information, Figure 5 became Figure 6 and Figure S4 became Figure S5.

Figure 4. Are the traces shown in this graph an average of several chambers? This is not mentioned in the caption. If they are indeed averages, some sort of indication of dispersion of data (e.g., standard deviation) would be useful, although it may be difficult to include too much data in the figure.

Yes, the traces shown in Figure 4 are the average of several chambers. As reviewer suggested, we mention it in the caption and we have also added the standard deviation to the traces shown.

Figure 7. Upper panel and lower panel content (drought vs pre-drought) is not consistent between the plot and the caption. Please correct.

We changed Figure 7 as mentioned in a previous comment and we corrected the caption accordingly.

Supporting information. The last sentence needs editing ("Samples were collected at The inert coated...").

We edited the sentence as suggested.

Reviewer #2 (Remarks to the Author):

In this article, the authors describe the effect of prolonged drought and rewetting on soil VOC fluxes in controlled conditions. The authors describe that under wet conditions, rainforest soil acts as a net sink of VOCs, in particular for isoprenoids, carbonyls and alcohols. However, this capacity decreases with an increase in drought conditions and at a certain level, the soil becomes the source of VOCs, which was related to soil microbial activity. The overall outcome of the study is not novel in a way that similar studies have been reported earlier (i.e., Monard et al. 2021, Gray et al. 2014, Trowbridge et al. 2020, Raza et al. 2017); however, this study is relatively more comprehensive and describes dynamics of some VOCs fate in soil under drought and rewetting condition. There are several main factors that were overlooked while performing this study which reduces its comprehensiveness.

It's all about soil uptake and emission but no analysis was conducted for soil VOCs, and some conclusions were speculated based on the patterns of the emission of VOCs without really describing whether it is the microbial activity or soil physiochemical characteristics playing a role. Microbes are stated as the main contributor to soil sink activity but no true evidence is reported. In addition, inducing soil microbial activity by adding a foreign substrate does not really reflect the soil process under natural conditions but can only aid in conclusions.

Some VOCs showed increased emission just after rain events indicating the role of soil properties in it. Soil physiochemical properties, hygroscopicity of soil, pore size, clay particles, organic matter, water holding capacity? These all have an important role in the retention of VOCs in soil, especially organic matter adsorbs more VOCs under wet conditions while clay particles adsorb more VOCs under dry conditions (Ong and Lion, 1991).

Thanks for your comments and for recognizing the comprehensiveness of this study compared to previous works. Soils are complex ecological systems where multiple chemical, physical and biological processes concur and interact in a way not easy to decipher. Several previous studies have attempted to elucidate the origin of uptake or emission of VOCs by soil. However, these studies were usually conducted under controlled conditions in which soil was removed from its natural conditions, and the conclusions drawn still had a large degree of uncertainty when contextualized in a real field scenario. For instance, the investigations on soil microorganisms were performed under laboratory conditions to avoid interferences on VOC fluxes by the heterogeneity of soil samples, differences in microbial community, and variability of soil properties (Cleveland & Yavitt, 1997). On the other hand, studies investigating soil properties were performed on lab prepared soils to avoid interferences on VOC fluxes by soil microorganisms (Ong & Lion, 1991; Serrano & Gallego, 2006). In the present study, we exploited the unique features of the tropical rainforest mesocosm of the Biosphere 2 to investigate the effects of prolonged drought and recovery on the soil VOC fluxes in situ without disturbing the soils. We performed a long-term experiment (about 4 months) collecting data with high frequency from 12 replicate chambers. Soil perturbations were minimized while the only environmental condition changed was the soil moisture level. Simultaneously to VOC fluxes from soil, we also monitored soil respiration, which is an index of soil microbial activity. As soil uptake capacity of isoprenoid compounds decreased in a similar fashion to respiration, this represents clear evidence that the consumption of these compounds was mainly due to the soil microbial activity. In addition, with the position specific ¹³C-labeled pyruvate experiments we clearly demonstrated the ability of soil microbes to produce VOCs and that this ability changes in wet or in dry soil.

The reviewer raised some concerns on the addition of ¹³C-pyruvate substrate to the soil as it does not reflect the soil process under natural conditions. However, pyruvate is a central metabolite with

high turnover that appears in soils naturally. The high potential of using position-specific ^{13}C -labeled pyruvate isotopologues as metabolic tracers to determine qualitative aspects of carbon flux patterns through metabolic pathways has already been demonstrated and exploited both for the soil microbial community (Dijkstra *et al.*, 2011) and for plants (Kreuzwieser *et al.*, 2021, Werner *et al.*, 2020). Moreover, we want to point out that the ^{13}C -labeled pyruvate experiments were conducted on isolated portions of soil measured only for few days and therefore we exclude that pyruvate experiments could have somehow affected the VOC fluxes dynamics in response to the drought. We agree with the reviewer that the manuscript lacked soil physicochemical properties. Therefore, we now provide additional soil physicochemical properties data (Table 2). To examine whether the soil physicochemical properties could explain the observed soil VOC fluxes over the rain rewet, we conducted partial least square regression (PLSR) analysis. PLSR is a multivariate technique used to predict a Y variable (VOC fluxes) with a number of X variables (soil physicochemical properties), which can be correlated. In addition to the soil physicochemical properties reported in Table 2, soil moisture (volumetric water content), soil matric potential (soil water availability to plants) and soil respiration measured from the 4 sites were also included as predictors in the PLSR analysis. For each site, averaged soil VOC emissions and averaged soil VOC uptake rates over the last two days of drought and over the first seven hours of rewet were used for PLSR analysis. Soil fluxes for each individual VOC from the four sites of the B2 TRF over the last two days of severe drought and over the first seven hours of the rain rewet are shown in Figure A. Statistical differences between soil VOC fluxes from each site were obtained from Tukey mean comparison test. Results from PLSR analysis shows that soil VOC emission and uptake were positively correlated with soil moisture, soil matric potential, and soil respiration and negatively correlated with soil clay content (Figure B). The higher the soil water content was during the rain rewet, the higher the release in the ambient air of VOCs (e.g., carbonyls) accumulated in the soil micropores and, at the same time, the abiotic dissolution of the ambient VOCs (e.g., alcohols) was higher. Additionally, the increased soil water content after rewet also increased the microbial activity that led to a higher VOC uptake (e.g., isoprenoids) and to a higher VOC release (e.g., dimethyl sulfide and methanethiol). The effects of clay content and soil water content are interconnected. Higher soil water content is associated with lower VOC sorption capacity of the clay minerals due to its hydrophilic character (Ong & Lion, 1991). Moreover, when clay minerals are wetted, they can swell, decreasing porosity, VOC diffusion, and therefore VOC emission and uptake. This explains the negative effect of soil clay content on soil VOC emissions and uptakes over the rain rewet.

Table 2 Soil physicochemical properties at each site of the B2 TRF where soil flux chambers were placed.

Site	S1	S2	S3	S4
Texture	Loam	Loam	Loam	Silt Loam
% Clay, <2 μm	17.90 \pm 1.74	23.75 \pm 0.00	17.93 \pm 0.03	25.07 \pm 0.19
% Silt, 2-50 μm	46.13 \pm 3.41	36.25 \pm 0.00	44.80 \pm 0.15	51.37 \pm 0.73
% Sand, 50 - 2000 μm	35.99 \pm 4.96	39.24 \pm 2.88	37.27 \pm 0.18	23.57 \pm 0.91
% Gravel	19.79	15.12	26.36	19.06
Bulk Density, g/cm^3	1.38 \pm 0.20	1.40 \pm 0.21	1.21 \pm 0.29	1.40 \pm 0.15
Porosity	0.48 \pm 0.07	0.47 \pm 0.08	0.54 \pm 0.11	0.47 \pm 0.08
Total Carbon, $\mu\text{g}/\text{mg}$	24.10 \pm 8.02	27.00 \pm 3.06	29.40 \pm 7.30	21.44 \pm 6.51
Total Nitrogen, $\mu\text{g}/\text{mg}$	1.90 \pm 0.69	2.26 \pm 0.18	2.54 \pm 0.80	1.70 \pm 0.42
Electrical conductivity, $\mu\text{S}/\text{cm}$	409.10 \pm 130.7	771.30 \pm 102.4	644.00 \pm 277.90	257.10 \pm 77.80
pH	7.26 \pm 0.08	7.44 \pm 0.14	7.31 \pm 0.11	6.98 \pm 0.32
Water holding capacity, %	66.94 \pm 6.70	60.48 \pm 3.38	59.03 \pm 2.82	70.78 \pm 4.85

Figure A Soil VOC fluxes from the four different sites of the B2 TRF over the last two days of severe drought and over the first seven hours after the first rain rewet. For the severe drought period, VOC fluxes from all 3 chambers placed at each of the 4 sites were considered. For the rewet period, for S1, S2, and S3 only VOC fluxes from the 2 chambers subjected to the rain rewet were considered. The boxes represent 25% to 75% of the dataset with the circle dots and central lines indicating the mean and median values, respectively. The whiskers indicate the minimum and maximum data points. Statistical differences between soil fluxes from each site were obtained from Tukey mean comparison test which accounts for Family-wise error rate. Statistically significant differences are labeled with asterisks: * $p \leq 0.05$; ** $p \leq 0.01$; *** $p \leq 0.001$.

Figure B Regression coefficients of partial least squares regression (PLSR) models and Variable Importance (VIP) for the covariance between the measured soil variables at four sites of the B2TRF and averaged soil VOC emissions and averaged soil VOC uptake over the rain rewet. Positive regression coefficients indicate a positive relationship and negative ones a negative relationship. Variables with a VIP > 1 are considered important (filled circle) while variables with VIP < 1 are considered less important (open circles).

References:

- Cleveland, C. & Yavitt, J. Consumption of atmospheric isoprene in soil. *Geophysical Research Letters - GEOPHYS RES LETT* 24, 2379–2382 (1997).
- Ong, S. K. & Lion, L. W. Trichloroethylene Vapor Sorption onto Soil Minerals. *Soil Science Society of America Journal* 55, 1559–1568 (1991).
- Dijkstra, P. et al. Probing carbon flux patterns through soil microbial metabolic networks using parallel position-specific tracer labeling. *Soil Biology and Biochemistry* 43, 126–132 (2011).
- Kreuzwieser, J. et al. Drought affects carbon partitioning into volatile organic compound biosynthesis in Scots pine needles. *New Phytol* 232, 1930–1943 (2021).
- Werner, C., Fasbender, L., Romek, K. M., Yáñez-Serrano, A. M. & Kreuzwieser, J. Heat Waves Change Plant Carbon Allocation Among Primary and Secondary Metabolism Altering CO₂ Assimilation, Respiration, and VOC Emissions. *Frontiers in Plant Science* 11, (2020).
- Serrano, A. & Gallego, M. Sorption study of 25 volatile organic compounds in several Mediterranean soils using headspace–gas chromatography–mass spectrometry. *Journal of Chromatography A* 1118, 261–270 (2006).

We bring the reviewer's points into the manuscript by including the following text.

Changes in the manuscript:

3.2 Rewet dynamics

..... To examine whether the soil physicochemical properties measured at four different sites of the B2 TRF (Table 1) contributed to the soil VOC fluxes over the rain rewet, partial least square regression (PLSR) analysis was conducted. In addition to the soil physicochemical properties reported in Table 2, soil moisture (volumetric water content), soil matric potential (soil water availability to plants) and soil respiration measured from the four sites were also included as predictors in the PLSR analysis. For each site, averaged soil VOC emissions and averaged soil VOC uptake rates over the last two days of drought and over the first seven hours of rewet were used for PLSR analysis. Soil fluxes for each individual VOC from the four sites of the B2 TRF over the last two days of severe drought and over the first seven hours of the rain rewet are shown in Figure S4. Results from PLSR analysis shows that soil VOC emission and uptake were positively correlated with soil moisture, soil matric potential, and soil respiration, and negatively correlated with soil clay content (Figure 5). The higher the soil water content was during the rain rewet, the higher the release in the ambient air of VOCs (e.g., carbonyls) accumulated in the soil micropores and, at the same time, rates of the abiotic dissolution of the ambient VOCs (e.g., alcohols) was higher. Additionally, the increased soil water content after rewet also increased the microbial activity that led to a higher VOC uptake (e.g., isoprenoids) and to a higher VOC release (e.g., dimethyl sulfide and methanethiol). The effects of clay content and soil water content are interconnected. Higher soil water content is associated with lower VOC sorption capacity of the clay minerals due to its hydrophilic character (Ong & Lion, 1991). Moreover, when clay minerals are wetted, they can swell, decreasing porosity, VOC diffusion, and therefore both VOC emission and uptake. This explains the negative effect of soil clay content on soil VOC emissions and uptakes over the rain rewet.

Table 1 Soil physicochemical properties at each site of the B2 TRF where soil flux chambers were placed.

Site	S1	S2	S3	S4
Texture	Loam	Loam	Loam	Silt Loam
% Clay, <2 μm	17.90 \pm 1.74	23.75 \pm 0.00	17.93 \pm 0.03	25.07 \pm 0.19
% Silt, 2-50 μm	46.13 \pm 3.41	36.25 \pm 0.00	44.80 \pm 0.15	51.37 \pm 0.73
% Sand, 50 - 2000 μm	35.99 \pm 4.96	39.24 \pm 2.88	37.27 \pm 0.18	23.57 \pm 0.91
% Gravel	19.79	15.12	26.36	19.06
Bulk Density, g/cm ³	1.38 \pm 0.20	1.40 \pm 0.21	1.21 \pm 0.29	1.40 \pm 0.15
Porosity	0.48 \pm 0.07	0.47 \pm 0.08	0.54 \pm 0.11	0.47 \pm 0.08
Total Carbon, $\mu\text{g}/\text{mg}$	24.10 \pm 8.02	27.00 \pm 3.06	29.40 \pm 7.30	21.44 \pm 6.51
Total Nitrogen, $\mu\text{g}/\text{mg}$	1.90 \pm 0.69	2.26 \pm 0.18	2.54 \pm 0.80	1.70 \pm 0.42
Electrical conductivity, $\mu\text{S}/\text{cm}$	409.10 \pm 130.7	771.30 \pm 102.4	644.00 \pm 277.90	257.10 \pm 77.80
pH	7.26 \pm 0.08	7.44 \pm 0.14	7.31 \pm 0.11	6.98 \pm 0.32
Water holding capacity, %	66.94 \pm 6.70	60.48 \pm 3.38	59.03 \pm 2.82	70.78 \pm 4.85

Figure S4 Soil VOC fluxes from the four different sites of the B2 TRF over the last two days of severe drought and over the first seven hours after the first rain rewet. For severe drought period, VOC fluxes from all 3 chambers placed at each of the 4 sites were considered. For rewet period, for S1, S2, and S3 only VOC fluxes from the 2 chambers subjected to the rain rewet were considered. The boxes represent 25% to 75% of the dataset with the circle dots and central lines indicating the mean and median values, respectively. The whiskers indicate the minimum and maximum data points. Statistical differences between soil fluxes from each site were obtained from Tukey mean comparison test which accounts for Family-wise error rate. Statistically significant differences are labeled with asterisks: * $p < 0.05$; ** $p < 0.01$; *** $p < 0.001$.

Figure 5 Regression coefficients of partial least squares regression (PLSR) models and Variable Importance (VIP) for the covariance between the measured soil variables at four sites of the B2TRF and averaged soil VOC emissions and averaged soil VOC uptake over the rain rewet. Positive regression coefficients indicate a positive relationship and negative ones a negative relationship. Variables with a VIP > 1 are considered important (filled circle) while variables with VIP < 1 are considered less important (open circles).

4 Discussion

..... Soil clay content played an important role in determining soil VOC fluxes over the rain rewet. Soil VOC uptake and emission rates both decreased with increasing soil clay content. This is because the capacity of the clay minerals to sorb and release VOCs decreased with increasing soil water content due to their hydrophilic character and to their tendency to swell in wet conditions (Ong & Lion, 1998).....

5.4 ¹³C pyruvate labeling

To identify the origin of emitted VOCs, soil was labeled with position specific ¹³C₁-pyruvate and ¹³C₂-pyruvate. Pyruvate is a central metabolite with high turnover that appears in soils naturally and serves as substrate for primary and secondary metabolic pathways as the C₁-carbon position of pyruvate is decarboxylated while the remaining acetyl-CoA can be involved in VOC biosynthesis. The high potential of using position-specific ¹³C-labeled pyruvate isotopologues as metabolic tracers to determine qualitative aspects of carbon flux patterns through metabolic pathways has already been demonstrated and exploited either for soil microbial communities (Dijkstra *et al.*, 2011) and for plants (Kreuzwieser *et al.*, 2021; Werner *et al.*, 2020). ¹³C-pyruvate was added in 9 additional soil chambers located at site 1, site 2, and site 3 (three for each site) of B2 TRF, adjacent to soil chambers measured over the whole campaign. ¹³C-pyruvate experiments were performed during pre-drought from 11th to 23th September and during severe drought from 6th to 18th November.

5.5 Soil properties

Soil moisture, soil temperature and soil matric potential were measured every 15 minutes by means of two sensors (SMT100, Truebner GmbH, Neustadt, Germany; TEROS 21, Meter Group, Pullman, WA, USA) installed in 5 cm soil depth from the surface close to the soil chamber sites.

Soil texture was determined using the sedimentation method. Soil bulk density was determined from oven-dry (110 °C) weight of the undisturbed cores of known volume (3 cm in length, 5.7 cm in diameter, Soil moisture Equipment Corp., Goleta, California) and porosity was calculated from bulk density. Total carbon and nitrogen were determined by combustion using Shimadzu TOC-VCSH analyzer (with solid state module SSM-5000A, Columbia, MD). Soil pH and electrical conductivity was determined in 1:1 soil-water suspension with VWR symphony pH meter (Radnor, PA). For water holding capacity (WHC), 20 g of sieved soil were placed on a glass funnel with a Whatman 40 filter paper. 40 mL of water was added to the plugged funnel to saturate the soil, and after 2 h, the soils were allowed to drain for 6 h. The wet soil was placed on an oven at 105 °C for 48 h to obtain the dry weight of the soil. The WHC was calculated as:

$$WHC [\%] = \frac{(Weight\ wet\ soil - Weight\ dry\ soil)}{Weight\ dry\ soil} \times 100 \quad (4)$$

5.6 Statistical analysis

..... Statistical differences between soil fluxes from each site were obtained from Tukey mean comparison test which corrects for Family-wise error rate. Partial least square regression analysis (PLSR) was conducted to assess for covariance between the soil properties and averaged VOC fluxes over the rain rewet. PLSR is a multivariate technique used to predict a Y variable (VOC fluxes) with a number of X variables (soil physicochemical properties), which can be correlated. To give equal importance for all variables, the data were centered and all variables were auto-scaled to unit variance. The PLS fitting model was cross-validated using the leave-one-out method. Variables with Variable Importance (VIP) > 1 are considered important while variables with VIP < 1 are considered less important. Tukey mean comparison test and PLSR analysis were performed using OriginPro (Version 2021b, OriginLab Corporation, Northampton, MA, USA).

What about the uptake of VOCs by plants?

We are not completely clear what the reviewer intends with this question. If the reviewer refers to possible VOCs uptake by plants inside the soil chambers, as stated in the method section “**5.3 Soil Fluxes measurements**”, the 12 soil chambers were installed on vegetation-free bare soil as the main focus of the study was to investigate the effect of drought on VOC fluxes only from the forest soil. Therefore, we exclude that plants could have somewhat contributed to the observed VOC fluxes from closed soil chambers. However, if the reviewer refers to other experiments performed during the B2WALD campaign to measure VOC fluxes from specific plants, these were measured from leaf chambers and these data will be the topic of another paper from the B2WALD campaign.

We bring the reviewer’s points into the manuscript by including the following text.

Changes in the manuscript:

5.3 Soil fluxes measurements

The 12 chambers were placed on PVC-collars (\varnothing : 20 cm) installed at 2-3 cm depth at four different sites of the B2 TRF eight weeks before the start of the measurements. **When the soil collars were installed, vegetation and litter inside the collars were removed in order to prevent any impact on VOC fluxes from the forest soil. Subsequently, any leaves falling into the chambers were immediately removed.**

Does air velocity under natural conditions have some role to play

Windy conditions could potentially generate negative pressure excursions inside a closed chamber (Venturi effect) that could cause a mass flow of air from the soil into the chamber, leading to overestimation of soil gas flux. However, the Licor chambers we used are featured with a pressure vent system to maintain pressure equilibrium even under windy conditions which, however, never occur in B2 TRF. The vent system is patented by Licor (Furtaw *et al.*, U.S. Patent 7,856,899., 2010) and was first introduced by Xu *et al.*, 2006. We bring the reviewer's point by including the following text in the manuscript.

References:

Furtaw, M.D., McDermitt, D.K., and Xu, L., "Vent and soil flux measurement system," U.S. Patent 7,856,899, December 28, 2010

Xu, L. *et al.* On maintaining pressure equilibrium between a soil CO₂ flux chamber and the ambient air. *Journal of Geophysical Research: Atmospheres* 111, (2006).

Changes in the manuscript:

5.2 Experimental set-up

Soil VOC fluxes were measured continuously using a proton transfer time of flight mass spectrometer (PTR-ToF-MS-8000, Ionicon Analytik GmbH, Innsbruck, Austria) directly connected to the outflow of an automated soil flux system consisting of a LI-8100 infrared gas analyzer (IRGA; for CO₂ fluxes measurement), a LI-8150 16-port multiplexer (Licor Inc., Lincoln, NA, USA) and 12 dynamic soil flux chambers (LI 8100-104 Long-Term Chambers with opaque lids, Licor Inc.). The soil flux system is a closed dynamic system in which the ambient air sampled with the chamber lid open is then recirculated once the chamber lid is closed, and the soil flux measurement is based on the calculation of the compound concentration development in the recirculated ambient air. **The soil chambers are featured with a pressure vent system to maintain pressure equilibrium inside the chambers even under windy conditions (Xu *et al.*, 2006).**

Role of temperature is overlooked, which is not only directly related to climate change but also to microbial activity and retention properties of VOCs.

We agree with the reviewer about the importance of the soil temperature on microbial activity and also on VOCs retention on soil surface. However, as the main scope of the study was to assess the drought effects, we controlled the temperature inside the B2 TRF to avoid having a seasonal trend in temperature, since this would have confounded drought effects (Werner *et al.*, 2021). As shown in Figure 1, the soil temperature was almost stable throughout the campaign and therefore we exclude its influence on soil VOC flux changes induced by dramatic decrease in soil moisture. To clarify this, we added a sentence in the results and method sections.

Changes in the manuscript:

References:

Werner, C. et al. Ecosystem fluxes during drought and recovery in an experimental forest. *Science* 374, 1514–1518 (2021)

3.1 Long term soil VOC fluxes dynamics

As expected, the soil temperature was relatively stable throughout the campaign (21.5- 25.5°C) and it showed a low diel variation with nighttime temperatures on average 1.4 ± 0.4 °C lower than the daytime temperatures. Part of this stability was achieved through the use of heaters to maintain nighttime air temperature above 15 °C during the latter part of the campaign in order to avoid having a seasonal trend in temperature which would have confounded the drought signal (Werner *et al.*, 2021).

5.1 The B2 TRF mesocosm and controlled drought experiment

The soil flux measurements were conducted during the Water, Atmosphere, and Life Dynamics campaign (B2-WALD) from September 2019 to January 2020 during which 65 days of drought were induced. During the whole campaign, the temperature inside the ecosystem was controlled in order to avoid having a seasonal trend in temperature which would have confounded the drought signal (Werner *et al.*, 2021).

Microbial activity was thought to be responsible for pulses of some VOCs like Dimethyl disulfide, is it possible for microbes to produce pulses of any VOC just after rewetting in a short time or there are some other abiotic factors involved?

As shown in Figure 4, we observed an emission pulse for all carbonyl compounds namely acetaldehyde, acetone, butanone and pentanone after the manual rewet and after the first rewet, and for the two sulfur compounds namely methanethiol and dimethyl sulfide after all rewets. The carbonyls emission pulse occurred within 2-3 hours after rewet and it only lasted for 2 hours after which all carbonyls started to be taken-up by the soil. In contrast, the sulfur pulse occurred about 9 hours after the manual rewet and the first rain rewet, and about 4 hours after the second rain rewet and the emissions lasted for about 3 days. Considering these timings of the observed rewet dynamics, we agree with the reviewer on the abiotic origin of the fast and short pulse observed for carbonyl compounds. The carbonyls could be generated by the immediate water-induced mobilization of the soil organic carbon such as release of intracellular osmolytes accumulated by water-stressed microorganisms, to microbial cell lysis caused by osmotic shock, and to the physical disruption of soil aggregates protecting organic matter (Unger *et al.*, 2010; Navarro-Garcia *et al.*, 2012). In addition, as carbonyl soil production and emission increased during drought, the soil micropores could have been filled with these compounds, and when rain water entered the micropores the water molecules replaced those of the carbonyls causing their release into the ambient air. However, regarding the sulfur compounds, the timings of the pulse and the concurrent pulse in soil respiration are strong evidence of the microbial origin of the sulfur emission pulse after the rewet.

References:

Unger, S., Máguas, C., Pereira, J. S., David, T. S. & Werner, C. The influence of precipitation pulses on soil respiration – Assessing the “Birch effect” by stable carbon isotopes. *Soil Biology and Biochemistry* 42, 1800–1810 (2010).

Navarro-García, F., Casermeiro, M. Á. & Schimel, J. P. When structure means conservation: Effect of aggregate structure in controlling microbial responses to rewetting events. *Soil Biology and Biochemistry* **44**, 1–8 (2012).

We bring the reviewer's point by including the following text in the manuscript.

Changes in the manuscript:

3.2 Rewet dynamics

.....A pulse in CO₂ soil emissions was observed after all rewet events (Figure 4, black plots). This phenomenon is known as the “Birch effect” and has been attributed to a rewetting-induced mineralization of labile soil organic carbon pools. The increased availability of these organic substrates after the rewet is thought to be due to an increased release of intracellular osmolytes accumulated by water-stressed microorganisms, to microbial cell lysis caused by osmotic shock, and to the physical disruption of soil aggregates protecting organic matter. **An emission pulse was also observed for carbonyl compounds namely acetaldehyde, acetone, butanone and pentanone (Figure 4, light green plots) after the rewet events on 12th December, and for the sulfur compounds namely methanethiol and dimethyl sulfide (Figure 4, red plots) after all rewet events.** The carbonyls pulse was fast and short as it occurred within 2-4 hours after rewet and lasted for about 2 hours. This indicates that the carbonyl pulse was abiotic in origin and attributable to the immediate water-induced mobilization of the soil organic carbon such as the rapid release of cell osmolytes, cell lysis caused by osmotic shock, and physical disruption of the soil organic matter (Unger *et al.*, 2010; Navarro-Garcia *et al.*, 2012). In addition, as the production and emission of carbonyls by the soil increased during drought (Figure 2), the soil micropores could have been filled with these compounds, and when rain water entered the micropores the water molecules replaced those of the carbonyls causing their release into the ambient air. In contrast to the carbonyls, the pulse of the sulfur compounds was slow (occurring 9 hours after the manual and first rain rewet and 4 hours after the second rain rewet) and long (it lasted for about 3 days), and it was concurrent with and very similar to the soil respiration pulse. These are strong indicators of the biotic origin of the sulfur pulse, attributable to the water induced mobilization and mineralization of the soil organic sulfur pools, whereby large insoluble sulfur-containing organic molecules are reduced to smaller soluble sulfur containing molecules by soil microbes or by extracellular soil enzyme.

Any relationship of outcomes with VOCs mass or class or retention properties?

We are not completely clear what the reviewer intends with “retention properties”. However, we identified a relationship between the soil VOC fluxes and VOC chemical class. We identified four classes of compounds namely isoprenoids, carbonyls, alcohols and sulfurs, and we described and discussed the results by class. This VOCs grouping criteria is consistent over the manuscript and it is further highlighted in all figures as we used the same color for the traces of VOCs belonging to the same class, i.e blue for isoprenoids, green for carbonyls, dark green for alcohols, red for sulfurs.

How do leaves or debris cover in tropical forests contribute to soil VOC flux? This is the ignored portion of this study.

We thank the reviewer for his comment. As stated in a previous comment, we removed vegetation and litter from the soil collars to prevent their effects on VOC fluxes from the forest soil.

5.3 Soil fluxes measurements

The 12 chambers were placed on PVC-collars (\varnothing : 20 cm) installed at 2-3 cm depth at four different sites of the B2 TRF eight weeks before the start of the measurements. **When the soil collars were installed, vegetation and litter inside the collars were removed in order to prevent any impact on VOC fluxes from the forest soil. Subsequently, any leaves falling into the chambers were immediately removed.**

Number of VOCs identified in soil seems low in numbers.

With the PTR-TOF-MS we detected hundreds of m/z signals and all of which are potentially gas phase compounds of interest. All peaks were carefully considered and we included in the paper only compounds that showed discernable soil fluxes. However, we take the reviewer's point, and now add a sentence in the section 5.2 of the manuscript to better explain how the VOCs were selected. For completeness, we now also include soil fluxes of 3 additional sulfur compounds namely methanethiol (Figure B, red plot), $C_4H_{10}S$ (Figure B, orange plot) and C_3H_8OS (Figure B, gray plot).

Methanethiol was emitted by the soil during pre-drought, then slightly taken-up during drought periods and similarly to dimethyl sulfide, it showed two emission pulses immediately after the two rain events. Methanethiol in soil is mainly originating from the metabolism of sulfur-containing amino acids by microorganisms and from the methylation of hydrogen sulfide. Dimethyl sulfide is formed through the methylation of methanethiol, potentially explaining why the soil emission pulses observed for dimethyl sulfide were one order of magnitude higher than methanethiol (Higgins *et al.*, 2006).

Soil emission was also observed for the other two sulfur-containing compounds $C_4H_{10}S$ and C_3H_8OS . $C_4H_{10}S$ soil emissions progressively increased from the early-drought period reaching the maximum during severe drought, and decreased during the recovery period. In contrast, C_3H_8OS soil emission started during early-drought and remained constant over the whole drought period. After the first rain event C_3H_8OS soil emission first suddenly decreased and then increased up to their maximum before the second rain event. After the second rain event C_3H_8OS soil fluxes recovered to pre-drought levels. Soil emissions of $C_4H_{10}S$, tentatively identified as isopropyl methyl sulfide, have been previously reported but its origin is not well understood (Mancuso *et al.*, 2015; Meischner *et al.*, 2022). Mancuso *et al.*, suggested $C_4H_{10}S$ as a potential intermediate product of the dimethyl sulfide metabolic pathway. However, in the present study, soil fluxes of dimethyl sulfide and $C_4H_{10}S$ showed rather different temporal patterns, indicating that they do not originate from the same soil process. We hypothesize that $C_4H_{10}S$ production in soils could either be due to microbial activity or to secondary chemical reactions occurring in the soil or on the soil surface. The detection of C_3H_8OS from soils is new and could be tentatively identified as 2-methylthioethanol, which is an intermediate product of methionine salvage pathway by microbes (Sekowska *et al.*, 2022). As methionine production demands high energy (Sekowska *et al.*, 2022), water stressed soil microbes recycled it leading to higher 2-methylthioethanol soil emissions.

Figure C Time series of soil fluxes for methanethiol, $C_4H_{10}S$ and C_3H_8OS . Lines represent averaged fluxes over the all 12 chambers. The shaded areas indicate the standard deviation. Background colors indicate the different phases of the campaign: pre-drought (white, DoY 270-279), early drought (light gray, DoY 280-305), severe drought (dark gray, DoY 305-346), and recovery (light blue, DoY 346-369). The first drought-ending rain event occurred at the start of the recovery period on DoY 346, and the vertical blue line indicates the time of the second rain event (DoY 353).

References:

Higgins, M. J. et al. Cycling of Volatile Organic Sulfur Compounds in Anaerobically Digested Biosolids and its Implications for Odors. *Water Environment Research* 78, 243–252 (2006).

Mancuso, S. et al. Soil volatile analysis by proton transfer reaction-time of flight mass spectrometry (PTR-TOF-MS). *Applied Soil Ecology* 86, 182–191 (2015).

Meischner, M. et al. Soil VOC emissions of a Mediterranean woodland are sensitive to shrub invasion. *Plant Biology* 24, 967–978 (2022).

Sekowska, A., Ashida, H. & Danchin, A. Revisiting the methionine salvage pathway and its paralogues. *Microbial Biotechnology* 12, 77–97 (2019).

Changes in the manuscript:

Figure 2 Time series of soil VOC fluxes. Lines represent averaged fluxes over the all 12 chambers. The shaded areas indicate the standard deviation. Background colors indicate the different phases of the campaign: pre-drought (white, DoY 270-279), early drought (light gray, DoY 280-305), severe drought (dark gray, DoY 305-346), and recovery (light blue, DoY 346-369). The first drought-ending rain event occurred at the start of the recovery period, and the vertical blue line indicates the time of the second rain event (DoY 353).

Figure 3 Relationship between the averaged normalized soil VOC fluxes and soil moisture, over the whole period of the campaign. To give the same weight to all VOC fluxes, for each VOC, daily averaged soil fluxes were normalized to their absolute maximum. Dots represents the average of normalized fluxes over all VOCs shown in Figure 2, for each day of the campaign. Black lines indicate the segmented regression model with the moisture threshold indicated by the vertical red line. Shaded gray area indicate the 95 % confidence interval of the regression model. The box plot chart shows the distribution of threshold estimates based on 1000 bootstrapped samples.

Figure 4 Rewet dynamics for soil respiration and VOC soil fluxes after the manual rewet (a), the first rain rewet (b) and the second rain rewet (c). To make comparable the dynamics of different compounds, for each compound, soil fluxes were normalized to their respective absolute maximum. Lines represent averaged values over 3 chambers for manual rewet, over 9 chambers for first rain rewet, and over all 12 chambers for second rain rewet. The shaded areas indicate the standard deviation. The data gap on

the first rain rewet plots is due to the fact that during that time only the 3 manually rewetted chambers were measured with a high temporal resolution with the aim to capture the fast dynamics.

Figure 6 Diel cycle observed during pre-drought (DoY 270-279), early drought (DoY 280-305), severe drought (DoY 305-346), and recovery (DoY 346-369) for a) isoprene soil flux, b) isoprene ambient concentration, c) isoprene deposition velocity, d) methyl nitrite flux, e) $C_4H_{10}S$ flux, f) soil moisture, g) temperature, and h) soil matrix potential. The boxes represent 25% to 75% of the dataset. The filled square and central lines indicate the mean and median values, respectively. The whiskers indicate the minimum and maximum data points at 1.5 times the interquartile range. Filled diamond indicate the outliers.

Table S 2 List of tentatively identified VOCs that showed measurable soil fluxes.

(m/z) measured	(m/z) exact	Mass accuracy [ppm]	Protonated ion	Potential compound
33.0331	33.0335	-12.11	CH ₅ O ⁺	Methanol
45.0332	45.0335	-6.66	C ₂ H ₄ O ⁺	Acetaldehyde
47.0488	47.0491	-6.38	C ₂ H ₇ O ⁺	Ethanol
49.0126	49.0106	40.80	CH ₅ S ⁺	Methanethiol
59.0498	59.0491	11.85	C ₃ H ₇ O ⁺	Acetone/propanal
62.029*	62.0236	87.06	CH ₄ ONO ⁺	Methyl nitrite
63.0255	63.0263	-12.69	C ₂ H ₇ S ⁺	Dimethyl sulfide
69.07	69.0699	1.45	C ₅ H ₉ ⁺	Isoprene
71.0493	71.0491	2.81	C ₄ H ₇ O ⁺	Methacrolein/methyl vinyl ketone
73.0649	73.0648	1.37	C ₄ H ₉ O ⁺	Butanone/butanal
85.0657	85.0648	10.58	C ₅ H ₉ O ⁺	Ethyl vinyl ketone/cyclopentanone/pentanal
87.0809	87.0804	5.74	C ₅ H ₁₁ O ⁺	Pentanone/pentanal
91.0555	91.0576	-23.06	C ₄ H ₁₁ S ⁺	Isopropyl methyl sulfide
93.0368	93.0369	-1.07	C ₃ H ₉ OS ⁺	2-methylthioethanol
137.1320	137.1325	-3.65	C ₁₀ H ₁₇ ⁺	Monoterpenes

*subtraction of the contribution from the ¹³C isotopologues.

Table S 3 Blank fluxes for all investigated VOCs. Mean flux and SD represent the average and the standard deviation, respectively, over three replicate measurements.

VOC	Mean blank flux [$\mu\text{mol m}^{-2} \text{h}^{-1}$]	SD [$\mu\text{mol m}^{-2} \text{h}^{-1}$]
Isoprene	-0.20211	0.24616
C ₅ H ₈ O	0.00928	0.00442
Monoterpenes	0.00880	0.01153
Methacrolein	0.00272	0.00090
Acetone fluxes	0.05415	0.00484
Acetaldehyde	0.01805	0.00394
Butanone	0.00923	0.00714
Pentanone	0.01138	0.00326
Methanethiol	0.00990	0.00360
Dimethyl sulfide	-0.00100	0.01353
C ₄ H ₁₀ S	-0.00009	0.00567

C_3H_8OS	0.02643	0.00581
Methanol	-0.10493	0.03807
Ethanol	0.11547	0.10256
Methyl nitrite	0.00095	0.00422

Figure S 1 Time series of ambient VOCs concentrations measured from the chambers with open lids during the last minute of the pre-purge. Lines represent averaged concentrations measured over all 12 chambers. The shaded areas indicate the standard deviation. Background colors indicate the different phases of the campaign: pre-drought (white, DoY 270-279), early drought (light gray, DoY 280-305), severe drought (dark gray, DoY 305-346), and recovery (light blue, DoY 346-369). Background colors indicate the different phases of the campaign: pre-drought (white), early drought (light gray), severe drought (dark gray), and recovery (light blue). The first drought-ending rain event occurred at the start of the recovery period, and the vertical blue line indicates the time of the second rain event (DoY 353).

3. Results

3.1 Long term soil VOC fluxes dynamics

Soil fluxes were also detected for the sulfur containing compounds **methanethiol** and **dimethyl sulfide** (Figure 2, red plots). They both showed low and highly variable soil fluxes for most of the campaign with a slight soil uptake for methanethiol and slight soil emission for dimethyl sulfide during severe drought. Both sulfur compounds showed two emission pulses directly after the rain

events with the methanethiol emission pulse about one order of magnitude lower than dimethyl sulfide emission pulse. Methanethiol in soils originates mainly from the metabolism of sulfur-containing amino acids by microorganisms and from the methylation of hydrogen sulfide. Dimethyl sulfide is formed through the methylation of methanethiol, potentially explaining why the soil emission pulses observed for dimethyl sulfide were in general one order of magnitude higher than the emission pulse of methanethiol (Higgins *et al.*, 2006; Carrion *et al.*, 2017). The associated increase also observed in ambient concentrations of both sulfur compounds (Figure S1) after rain events clearly showed that soil can significantly contribute to local ambient concentrations of sulfur compounds.

Soil emissions were also observed for other two sulfur-containing compounds $C_4H_{10}S$ and C_3H_8OS . $C_4H_{10}S$ soil emissions progressively increased starting from the early-drought period reaching a maximum during severe drought and decreased back to pre-drought levels during the recovery period. In contrast, C_3H_8OS soil emission started during early-drought and remained constant over the whole drought period. After the first rain event, C_3H_8OS soil emission first suddenly decreased and then increased up to their maximum before the second rain event. After the second rain event C_3H_8OS soil fluxes recovered to pre-drought levels. Soil emissions of $C_4H_{10}S$, tentatively identified as isopropyl methyl sulfide, have been previously reported but its origin is not well understood (Mancuso *et al.*, 2015; Meischner *et al.*, 2022). Mancuso *et al.*, suggested $C_4H_{10}S$ as a potential intermediate product of the dimethyl sulfide metabolic pathway. However, in the present study, soil fluxes of dimethyl sulfide and $C_4H_{10}S$ showed rather different temporal patterns, indicating that they do not originate from the same soil process. We hypothesize that $C_4H_{10}S$ production in soils could either be due to microbial activity or to secondary chemical reactions occurring on soil surface. In contrast to $C_4H_{10}S$, the detection of C_3H_8OS from soil is new and could be tentatively identified as 2-methylthioethanol, which is an intermediate product of methionine salvage pathway by microbes (Sekowska *et al.*, 2022). As methionine production demands high energy (Sekowska *et al.*, 2022), water stressed soil microbes recycled it leading to higher 2-methylthioethanol soil emissions.....

.....Soil moisture was a key driver for VOC fluxes and the relationship between the average of normalized VOC fluxes and soil moisture was non-linear evolving around a soil moisture threshold of ~19%, as determined by segmented regression (Figure 3). Below this soil moisture threshold, the VOC uptake capacity of the soil dramatically decreased and the soil started to be a source of VOCs. This suggests that 19% represents the soil moisture threshold corresponding to the point when the water-stressed soil microbes started producing and accumulating protective osmolytes, including VOCs, to reduce their internal water potential to avoid dehydrating and dying.

3.2 Rewet dynamics

.....A pulse in CO_2 soil emissions was observed after all rewet events (Figure 4, black plots). This phenomenon is known as the “Birch effect” and has been attributed to a rewetting-induced mineralization of labile soil organic carbon pools. The increased availability of these organic substrates after the rewet is thought to be due to an increased release of intracellular osmolytes accumulated by water-stressed microorganisms, to microbial cell lysis caused by osmotic shock, and to the physical disruption of soil aggregates protecting organic matter. An emission pulse was also observed for carbonyl compounds namely acetaldehyde, acetone, butanone and pentanone (Figure 4, light green plots) after the rewet events on 12th December, and for the sulfur compounds namely, methanethiol and dimethyl sulfide (Figure 4, red plots) after all rewet events. The carbonyls pulse was fast and short as it occurred within 2-4 hours after the rewets and it lasted for about 2 hours. This indicates that the carbonyl pulse was abiotic in origin and attributable to the immediate water-induced mobilization of the soil organic carbon such as the rapid release of cell osmolytes, cell lysis

caused by osmotic shock, and physical disruption of the soil organic matter. In addition, as the production and emission of carbonyls by the soil increased during drought (Figure 2), the soil micropores could have been filled with these compounds, and when rain water entered the micropores the water molecules replaced those of the carbonyls causing their release into the ambient air. In contrast to the carbonyls, the pulse of the sulfur compounds was slow (occurred 9 hours after the manual and first rain rewet and 4 hours after the second rain rewet) and long (it lasted for about 3 days), and it was concurrent with and very similar to the soil respiration pulse. These are strong indicators of the biotic origin of the sulfur pulse, attributable to the water induced mineralization of the soil organic sulfur pools, whereby large insoluble sulfur-containing organic molecules are reduced to smaller soluble sulfur containing molecules by soil microbes or by extracellular soil enzyme.

As shown in Figure 1 for CO₂ and in Figure 2 for methanethiol and dimethyl sulfide, the emission pulses following the second rain event were significantly lower in absolute magnitude compared to the pulses following the first rain rewet, indicating that shorter drought-rewet cycles induce a lower mobilization and mineralization of the soil organic matter or that induce a lower build-up of substrate pools. Indeed, the subsequent rain events conducted every second day starting from 21 December did not induce any VOCs and CO₂ soil emission pulses.

The soil uptake rates of isoprene, C₅H₈O and monoterpenes considerably increased only the day after the rewets (Figure 4, blue plots), reflecting the time needed for the microbes responsible for the consumption of these compounds to restart their activity. In contrast, the uptake of MACR+MVK peaked within a few hours after the rewet events most probably due to its abiotic dissolution in wet soil. An increase in soil uptake of alcohols was observed within 4 hours after the rewets (Figure 4, dark green plots) as a consequence of the simultaneous increase in their ambient concentrations (Figure S1) and to their abiotic dissolution in wet soil. In response to all rewet events, C₃H₈OS soil emission considerably decreased and the soil switched to taking up C₃H₈OS, but after a few hours the emission was restored again. Methyl nitrite soil emissions (Figure 4, brown plots) slowly decreased in response to the rewet events likely as a consequence of decreasing HONO production with increasing soil moisture. C₄H₁₀S fluxes decreased in response to the rewets in a similar fashion of methyl nitrite suggesting that the soil emissions of these two compounds could have originated from similar processes.

3.3 Diel dynamics

.....In contrast, the soil emissions of carbonyl compounds as well as those of methanethiol, dimethyl sulfide and C₃H₈OS did not show any diel cycle. A diel cycle was also observed for isoprenoid deposition velocities (Figure 6c), methyl nitrite emissions (Figure 6d) and C₄H₁₀S emissions (Figure 6e) during the severe drought period.

Higher daytime emission of methyl nitrite compared to nighttime can be attributed to a higher HONO production during the day which has been attributed to a photo-enhanced conversion of NO₂ or nitrate photolysis on the soil. The diel cycle observed for C₄H₁₀S soil emissions support the hypothesis that it could have originated from near-ground chemical reactions or, similarly to methyl nitrite, from photo-enhanced reactions on soil surface.

4. Discussion

In normal wet conditions, the soil of the experimental rainforest acted as a net VOC sink. The soil uptake capacity progressively decreased in response to increasing drought and, under severe drought conditions the soil started to be a strong source of several VOCs, including carbonyls, methyl nitrite, $C_4H_{10}S$ and C_3H_8OS

..... The emission pulse observed for the two sulfur compounds, namely methanethiol and dimethyl sulfide after soil rewetting was similar to the Birch effect and was therefore attributable to the mineralization of the soil organic sulfur pools by soil microbes and enzymes. The increase also observed in ambient concentrations occurred simultaneously to the soil emission pulses of the two sulfur compounds, showing that soil can significantly contribute to local ambient concentrations of sulfur compounds. It should be noted that on the global scale, the ocean is a larger source of dimethyl sulfide than the rainforest (Wang *et al.*, 2018). However, due to the relatively short atmospheric lifetime of dimethyl sulfide in the tropics (ca. 1 day) and the stronger convection experienced overland, rainforest emissions can still be important to local and regional chemistry. Dimethyl sulfide is of high relevance in atmospheric chemistry as it can be oxidized to sulfuric acid, contribute to new particle formation, and ultimately grow to form cloud condensation nuclei. The emission of the additional sulfur-containing compound, i.e., methanethiol, would strengthen this effect.....

.....Soil fluxes of several VOCs followed a diel cycle with higher emission and uptake rates both occurring during daytime compared to nighttime. Soil uptake rates of isoprenoids closely followed the diel cycle of their atmospheric concentrations, while diel cycles in methyl nitrite and $C_4H_{10}S$ soil emissions were a consequence of light dependent processes at or near to the soil surface.....

5.2 Experimental set-up

.....Only compounds that showed discernable soil fluxes were considered for further analysis. These compounds are reported in Table S2, along with tentative identifications for the underlying VOC species based on previous literature.....

VOC production is highly sensitive to the nutrient's sources, does the addition of ^{13}C -labelled pyruvate hold the merit of the natural process of soil?

Thanks for your comment. As mentioned above, pyruvate is not a nutrient source but a central metabolite with high turnover that appears in soils naturally. The high potential of using position-specific ^{13}C -labeled pyruvate isotopologues as metabolic tracers to determine qualitative aspects of carbon flux patterns through metabolic pathways has already been demonstrated and exploited for soil microbial community (Dijkstra *et al.*, 2011) and for plants (Kreuzwieser *et al.*, 2021; Werner *et al.*, 2020). Moreover, we would like to point out that the ^{13}C -labeled pyruvate experiments were conducted on isolated portions of soil measured only for a few days and therefore we can exclude that the pyruvate experiments could have somehow affected the VOC fluxes observed over the whole campaign.

References:

Dijkstra, P. et al. Probing carbon flux patterns through soil microbial metabolic networks using parallel position-specific tracer labeling. *Soil Biology and Biochemistry* 43, 126–132 (2011).

Kreuzwieser, J. et al. Drought affects carbon partitioning into volatile organic compound biosynthesis in Scots pine needles. *New Phytol* 232, 1930–1943 (2021).

Werner, C., Fasbender, L., Romek, K. M., Yáñez-Serrano, A. M. & Kreuzwieser, J. Heat Waves Change Plant Carbon Allocation Among Primary and Secondary Metabolism Altering CO₂ Assimilation, Respiration, and VOC Emissions. *Frontiers in Plant Science* **11**, (2020).

We bring the reviewer's point by adding the following text to the manuscript.

Changes in the manuscript:

5.4 ¹³C pyruvate labeling

To identify the origin of emitted VOCs, soil was labeled with position specific ¹³C₁-pyruvate and ¹³C₂-pyruvate. Pyruvate is a central metabolite with high turnover that appears in soils naturally and serves as substrate for primary and secondary metabolic pathways as the C₁-carbon position of pyruvate is decarboxylated while the remaining acetyl-CoA can be involved in VOC biosynthesis. The high potential of using position-specific ¹³C-labeled pyruvate isotopologues as metabolic tracers to determine qualitative aspects of carbon flux patterns through metabolic pathways has already been demonstrated and exploited either for soil microbial communities (Dijkstra *et al.*, 2011) and for plants (Kreuzwieser *et al.*, 2021; Werner *et al.*, 2020). ¹³C-pyruvate was added in 9 additional soil chambers located at site 1, site 2, and site 3 (three for each site) of B2 TRF, adjacent to soil chambers measured over the whole campaign. ¹³C-pyruvate experiments were performed during pre-drought from 11th to 23th September and during severe drought from 6th to 18th November.

Minor issues

Starting title with 'The' is not appropriate and also the use of the abbreviation 'VOC'.

Thank you. We changed the title in "Effects of prolonged drought and recovery on volatile organic compound fluxes from rainforest soil"

Line 36-38: That is an outdated statement

Thank you. We deleted the sentence

Line 43-45: Is there any role soil minerals or particles (i.e., clay) play in this process?

The possible role of the soil particles as abiotic drivers of soil VOC fluxes is already mentioned in the sentence to which the reviewer refers. However, we consider the reviewer's points in the manuscript by including an additional reference about the role of clay minerals (Ong and Lion, 1991).

References:

Ong, S. K. & Lion, L. W. Trichloroethylene Vapor Sorption onto Soil Minerals. *Soil Science Society of America Journal* 55, 1559–1568 (1991).

Changes in the manuscript:

Introduction

.....while the abiotic processes include dissolution **into** or evaporation from soil water, adsorption **onto or desorption from** soil particles, reaction with soil chemicals, and evaporation from leaf litter (Ong & Lion, 1991).....

Line 52-53: What about temperature rise? It is an important component of climate change.

Thanks for your comment. We agree that the temperature is an important component of climate change. We modified the sentence as follows.

Changes in the manuscript:

Introduction

....Among the predicted impacts of climate change, **temperature** and drought frequency and duration are expected to increase worldwide...

Line 55: rainforest contribution 70%? Soil or plant or both?

We were referring to the total ecosystem emissions, but we agree with the reviewer that the statement can cause confusion to the reader. Therefore, we clarified this by modifying the sentence.

Changes in the manuscript:

Introduction

This is particularly relevant for tropical rainforests, **as it is estimated that emissions from these ecosystems represent about 70% of the total source of biogenic VOCs to the atmosphere.**

Line 110: Abiotic dissolution is underestimated term for VOCs fate in the soil overlooking soil physical properties' role in the absorption or adsorption of VOCs in soil.

Thanks for your comment. As stated above, we have now added a comprehensive suite of soil physicochemical properties and we have discussed their relation to the observed VOC soil fluxes.

Line 161: "19% represents the soil moisture". This conclusion is not very important without exploring soil properties contribution.

Thanks for your comment. As stated above, we now report additional soil physicochemical properties and have discussed their relation to the observed VOC soil fluxes.

Line 224-226: What could be the reason for the depletion of isoprenoids in ambient air at night?

Plant emissions of isoprenoids are metabolically linked to plant photosynthesis (i.e., light driven) with higher emission at higher photosynthetic rates (Byron *et al.*, 2022). As a result, ambient concentrations of isoprenoids are high during daylight and low during night.

References:

Byron, J. *et al.* Chiral monoterpenes reveal forest emission mechanisms and drought responses. *Nature* 609, 307–312 (2022).

We bring the reviewer's point by adding the following text to the manuscript.

Changes in the manuscript:

3.3 Diel dynamics

The decrease observed in isoprenoid deposition velocity during the night was most probably due to substrate limitation in a very depleted ambient air at night (Figure 5b). **This is because plant emissions of isoprenoids are metabolically linked to plant photosynthesis with higher emission at higher photosynthetic rates (Byron *et al.*, 2022). As a result, ambient concentrations of isoprenoids during nighttime are extremely low.**

line 273: sudden use of an abbreviation

Thanks. We corrected it.

Line 308: rewet by adding ~2.2 L (~22.5 mm) of water per chamber? does the sudden application of water affect osmotic shock or VOC emission properties?

Thanks for this comment. The amount of water added to the three manual chambers was the same that the rest of the forest received from the overhead sprinklers 05:30 hours later during the rain rewet. The sprinklers were set at their normal capacity as during the pre-drought period. We agree with the reviewer that the water addition to the dry soil induced an osmotic shock that caused a rapid release of cellular osmolytes and cell lysis with consequent emission of VOCs. In particular, as discussed above, the rewet dynamics shown in Figure 4, strongly indicate that the fast and short emission pulse observed for carbonyl compounds was mainly attributable to the immediate water-induced mobilization of the soil organic carbon such as the rapid release of cell osmolytes, cell lysis caused by osmotic shock, and physical disruption of the soil organic matter.

Changes in the manuscript:

3.2 Rewet dynamics

.....A pulse in CO₂ soil emissions was observed after all rewet events (Figure 4, black plots). This phenomenon is known as the "Birch effect" and has been attributed to a rewetting-induced mineralization of labile soil organic carbon pools. The increased availability of these organic substrates after the rewet is thought to be due to an increased release of intracellular osmolytes accumulated by water-stressed microorganisms, to microbial cell lysis caused by osmotic shock, and to the physical disruption of soil aggregates protecting organic matter. **An emission pulse was also observed for carbonyl compounds namely acetaldehyde, acetone, butanone and pentanone (Figure 4, light green plots) after the rewet events on 12th December, and for the sulfur compounds, namely methanethiol and dimethyl sulfide (Figure 4, red plots) after all rewet events. The carbonyls pulse was fast and short as it occurred within 2-4 hours after the rewets and it lasted for about 2 hours. This indicates that the carbonyl pulse was abiotic in origin and attributable to the immediate water-induced mobilization of the soil organic carbon such as the rapid release of cell osmolytes, cell lysis caused by osmotic shock, and physical disruption of the soil organic matter. In addition, as the production and emission of carbonyls by the soil increased during drought (Figure 2), the soil micropores could have been filled with these compounds, and when rain water entered the micropores the water molecules replaced those of the carbonyls causing their release into the ambient air. In contrast to the carbonyls, the pulse of the sulfur compounds was slow (occurred 9**

hours after the manual and first rain rewet and 4 hours after the second rain rewet) and long (it lasted for about 3 days), and it was concurrent with and very similar to the soil respiration pulse. These are strong indicators of the biotic origin of the sulfur pulse, attributable to the water induced mineralization of the soil organic sulfur pools, whereby large insoluble sulfur-containing organic molecules are reduced to smaller soluble sulfur containing molecules by soil microbes or by extracellular soil enzyme.....

Line 345: How many VOCs are in that mixture?

We used two gas cylinders containing two different VOC mixtures in order to allow explicit calibration of a wide range of VOCs. Details on the VOC mixtures we used for PTR-ToF-MS calibration are reported in Table S1. We modified the method section making this point clearer to the reader.

Changes in the manuscript:

5.2 Experimental set-up

...Nocturnal calibrations, starting from midnight, were performed using a standard gas cylinder containing different multi-VOC component calibration mixture in Ultra-High Purity (UHP) nitrogen (Apel-Riemer Environmental, Inc., Colorado, USA). Two calibration standard cylinders were used during the campaign to allow explicit calibration of a wide range of VOCs. The first cylinder was used for two periods: from 18 September 2019 to 6 November 2019; and from 17 December 2019 to 20 January 2020. The second cylinder was used from 7 November 2019 to 16 December 2019. VOC gas standards included in the two calibration standard cylinders with their respective detection limit (LOD) and total uncertainty are reported in Table S1. For daily calibration the VOC mixture was subjected to 5-step dynamic dilutions by means of a liquid calibration unit (LCU, IONICON Analytik, Innsbruck, Austria). The gas standard was equilibrated in the LCU for one hour prior to the start of calibration. The zero-air flow was held constant at 1000 sccm, while the gas standard flow was changed every 15 min starting from 40 sccm until 0 sccm in 10 sccm steps. To calibrate at the same humidity level observed in the B2 TRF, 20 $\mu\text{L}/\text{min}$ of milli-Q water were dynamically nebulized into the evaporation chamber of the LCU.....

Line 296: The enclosed air is therefore relatively rich in primary VOC emissions and relatively poor in oxidized products. So, do natural conditions already compromised?

Thanks for your comment. The B2 TRF model ecosystem demonstrated similar behavior to the world's tropical rainforests and it allowed the study tropical ecosystem responses to environmental changes (Rascher *et al.*, 2004, Pegoraro *et al.*, 2006). The low concentration of atmospheric oxidants, namely ozone and hydroxyl radicals, represent one of the most important features of the B2 TRF mesocosm as it allows the estimation of the exchange rate of highly reactive VOCs as the ambient VOC concentration reflects the ecosystem VOC dynamics.

References:

Rascher, U. *et al.* Functional diversity of photosynthesis during drought in a model tropical rainforest – the contributions of leaf area, photosynthetic electron transport and stomatal conductance to reduction in net ecosystem carbon exchange. *Plant, Cell & Environment* **27**, 1239–1256 (2004).

Pegoraro, E., Rey, A., Abrell, L., Van Haren, J. & Lin, G. Drought effect on isoprene production and consumption in Biosphere 2 tropical rainforest. *Global Change Biology* **12**, 456–469 (2006)..

We agree with the reviewer that the sentence could lead to a misinterpretation and therefore we changed it as follow.

Changes in the manuscript:

5.1 The B2 TRF mesocosm and controlled drought experiment

The B2 TRF mesocosm is a fully enclosed ecosystem which allows temperature, humidity, atmospheric gas composition, and precipitation to be manipulated. **The B2 TRF mesocosm demonstrated similar behavior to the world's tropical rainforests and it allows to study tropical ecosystem responses to environmental changes (Rascher *et al.*, 2004, Pegoraro *et al.*, 2006). The mesocosm has an area of 1940 m² and a volume of 26700 m³ and the vegetation is rooted in 2-4 m of soil.** The low ozone (O₃) concentration (ca. 1 ppbV) and the absence of hydroxyl radical (OH) formation inside the B2 TRF due the UV-light filtering by the glass, prevent VOC oxidation **allowing the estimation of the fluxes of highly reactive VOCs since the ambient VOC concentrations reflect the ecosystem VOC dynamics.**

Reviewer #3 (Remarks to the Author):

The manuscript by Pugliese and co-authors explore soil VOCs fluxes during drought and during post drought recovery in an experimental rainforest. The manuscript is well in line with the quite recent interest from the community on soil VOCs emissions. The manuscript is well written, and the results are well presented. In particular, the authors observed emissions pulse of dimethyl sulfide after soil rewatering, and emissions of methyl nitrite under very severe drought conditions. These are interesting results as these two compounds are usually associated with oceanic emissions and are highly reactive in the atmosphere. To my knowledge, this is the first time that soil methyl nitrite emissions are observed.

Thanks for this positive feedback and for highlighting the novelty of the results and their relevance for the community on soil VOC fluxes. We address your comments below.

However, the authors failed to convince me that their results imply a significant impact of soil VOC on atmospheric chemistry and climate and the relative gain of adding soil VOC in land surface models:

lines 29-30 'Results show that, the extended drought periods predicted for tropical rainforest regions will strongly affect soil VOC fluxes thereby impacting atmospheric chemistry and climate'. Lines 284-288 'Prolonged drought and recovery had a major impact on soil VOC fluxes from the experimental rainforest, affecting the composition and quantity of VOCs in the atmosphere of the enclosed ecosystem. Soil VOC fluxes and their parametrization related to soil moisture levels must be included in atmospheric models to simulate current atmospheric chemistry and to improve climate model predictions of ecosystem responses to drought'. Indeed, the authors found a pulse of Dimethyl sulfide about 0.2 $\mu\text{mol}/\text{m}^2/\text{h}$ lasting for less than 10 days after rewatering. It seems relatively small when compared to oceanic fluxes which annual mean varies roughly between 0.15 and 0.35 $\mu\text{mol}/\text{m}^2/\text{h}$ and last all year around (cf. Wang, S., Maltrud, M., Elliott, S. et al. Influence of dimethyl sulfide on the carbon cycle and biological production. *Biogeochemistry* 138, 49–68 (2018). <https://doi.org/10.1007/s10533-018-0430-5>).

We agree with the reviewer that viewed globally the ocean is a far greater overall source of dimethyl sulfide than the rainforest. This point we now include by citing the paper suggested by the reviewer. However, given the relatively short atmospheric lifetime of dimethyl sulfide (ca. 1 day in the tropics) local rainforest sources can still be of relevance, regionally and through convection. Distant ocean sources of dimethyl sulfide will simply not reach the central rainforest.

Changes in the manuscript:

Discussion

.....The increase also observed in ambient concentrations occurred simultaneously to the soil emission pulses of the two sulfur compounds, showing that soil can significantly contribute to local ambient concentrations of sulfur compounds. It should be noted that on the global scale, the ocean is a larger source of dimethyl sulfide than the rainforest (Wang *et al.*, 2018). However, due to the relatively short atmospheric lifetime of dimethyl sulfide in the tropics (ca. 1 day) and the stronger convection experienced overland, rainforest emissions can still be important to local and regional chemistry. Dimethyl sulfide is of high relevance in atmospheric chemistry as it can be oxidized to sulfuric acid, contribute to new particle formation, and ultimately grow to form cloud condensation nuclei. The emission of the additional sulfur-containing compound, i.e., methanethiol, would strengthen this effect.....

The same applies for methyl nitrite. Maximum of soil emissions showed in the manuscript corresponds to annual mean flux in equatorial oceans (Fisher, J. A., Atlas, E. L., Barletta, B., Meinardi, S., Blake, D. R., Thompson, C., et al. (2018). Methyl, ethyl, and propyl nitrates: Global distribution and impacts on reactive nitrogen in remote marine environments. *Journal of Geophysical Research: Atmospheres*, 123, 12,429– 12,451.

<https://doi.org/10.1029/2018JD029046>). However, this maximum of emissions are found under extreme drought conditions, corresponding to a reduction of more than 50% of soil moisture (Figure 1). This reduction of soil moisture must be put in context. As the authors stated line 256-258, soil moisture anomaly were of almost 30% during the strong El Niño drought in 2015/2016. We are therefore still far away from a reduction of 50% of soil moisture, which seems to me very close to the permanent wilting point. To address the importance of their findings in term of impact, I wish the authors would have compared their findings with what we know from oceanic fluxes studies. That might have moderated (or not) their conclusion of the necessity to incorporate soil VOC fluxes into land surface models. Consequently, it is difficult for me to assess if the manuscript is relevant for Nature Communications.

There appears to have been some confusion concerning the compound we report here (and others see Stevenson *et al.*, 1964, Magalhães *et al.*, 1987) – methyl nitrite (CH₃ONO), and the compounds mentioned in the reference given by the reviewer which are nitrates (e.g., CH₃ONO₂). The nitrite we report is unstable with respect to photolysis, rapidly forming NO and formaldehyde (lifetime ca. 2 minutes, Taylor *et al.*, 1980). In the atmospheric boundary layer this additional NO will enhance the concentration of OH radicals through the reaction NO + HO₂ → OH + NO₂. Furthermore, photolysis of formaldehyde also yields OH radicals. Anything that enhances the OH radical abundance can have a knock-on effect and shorten the lifetimes of toxic (e.g., CO) and radiative gases (e.g., CH₄). We contend that it is of interest to assess the size of this effect in models.

The alkyl nitrates mentioned by the reviewer are formed in the atmosphere as minor products of the $\text{RO}_2 + \text{NO}$ reaction, and they are also formed in surface seawater (see Chuck *et al.*, 2002). However, their atmospheric lifetime is much longer as they do not photolyze rapidly (lifetime circa 1-2 weeks, Clemitshaw *et al.*, 1997) or react rapidly with OH. We did not detect or measure the alkyl nitrates.

References:

Taylor, W. D. *et al.* Atmospheric photodissociation lifetimes for nitromethane, methyl nitrite, and methyl nitrate. *International Journal of Chemical Kinetics* 12, 231–240 (1980).

Stevenson, F. J. & Swaby, R. J. Nitrosation of Soil Organic Matter: I. Nature of Gases Evolved During Nitrous Acid Treatment of Lignins and Humic Substances. *Soil Science Society of America Journal* 28, 773–778 (1964).

Magalhães, A. M. T. & Chalk, P. M. Factors affecting formation of methyl nitrite in soils. *Journal of Soil Science* 38, 701–709 (1987).

Chuck, A. L., Turner, S. M. & Liss, P. S. Direct Evidence for a Marine Source of C1 and C2 Alkyl Nitrates. *Science* 297, 1151–1154 (2002).

Clemitshaw, K. C. *et al.* Gas-phase ultraviolet absorption cross-sections and atmospheric lifetimes of several C2–C5 alkyl nitrates. *Journal of Photochemistry and Photobiology A: Chemistry* 102, 117–126 (1997).

Changes in the manuscript:

Discussion

Methyl nitrite emissions are highly relevant for the atmospheric chemistry in the boundary layer due to its rapid photolysis to NO and formaldehyde (Taylor *et al.*, 1980). Both compounds serve to elevate concentrations of OH radicals, the atmosphere's primary oxidant. Since methyl nitrite emissions occurred only during drought, the presented results suggest that future climate change associated drought periods will also affect diel rainforest carbon cycles and atmospheric oxidation chemistry.

Reviewer #2 (Remarks to the Author):

The authors have answered all of my queries. Generally, manuscript is well written and results description is satisfactory. I have a couple of comments.

Maybe should add more method details (a couple of sentences) in last paragraph of the introduction. Otherwise need to move to methods section before reading results section.

Is it not better to correlate clay content and other soil properties with each of VOC whose dynamics are described in the manuscript, as all VOCs have different retention properties and some have distinct patterns. Comparison with average VOC uptake and emission overlooks interesting details.

Organic matter is an important VOC adsorber in soil, Organic matter contents? As another important soil adsorber, clay content showed a significant effect on VOCs dynamics in soil even seems higher than moisture content parameters in Fig. 5.

And this might also answer partially to statements like line 129

Reviewer #3 (Remarks to the Author):

I am happy with the revisions and don't have additional comments.

Dear Reviewers,

Thank you very much for your time and for recommending our manuscript for publication. We are happy to present the final version of the manuscript, revised according to your comments. These comments were extremely helpful during the first revision, and we are grateful for this chance to further improve and now finalize the paper by taking the last remaining comments into account.

In the blue text below, we give first a general discussion of the main points raised and then point-by-point discussions of all the reviewer's questions noted in bold and black. Unmodified text of the manuscript is reported in plain black and all changes to the manuscript are marked in red.

Sincerely,

Dr. Giovanni Pugliese

Corresponding author

REVIEWERS' COMMENTS

Reviewer #2 (Remarks to the Author):

The authors have answered all of my queries. Generally, manuscript is well written and results description is satisfactory. I have a couple of comments.

Thank you for this positive feedback. All points raised are addressed in detail below.

Maybe should add more method details (a couple of sentences) in last paragraph of the introduction. Otherwise need to move to methods section before reading results section.

As suggested, we added more method details in the last paragraph of the introduction.

Changes in the manuscript:

2. Introduction

.....In this study, we conducted a long-term drought experiment (B2-WALD campaign) in the enclosed experimental Biosphere 2 Tropical Rainforest (B2 TRF, Arizona, USA), to assess the effects of prolonged and severe drought followed by rewetting on soil VOC fluxes direction and magnitude. The soil VOC fluxes were monitored continuously and in real-time by means of a proton-transfer-reaction time-of-flight mass spectrometer (PTR-ToF-MS) connected to 12 closed dynamic soil chambers placed in 4 different sites of the B2 TRF.....

Is it not better to correlate clay content and other soil properties with each of VOC whose dynamics are described in the manuscript, as all VOCs have different retention properties and some have distinct patterns. Comparison with average VOC uptake and emission overlooks interesting details.

Thanks for your comment. As suggested, we now performed PLSR analysis grouping the VOCs by compound class. Isoprenoid compounds include isoprene, C₅H₈O, MACR+MVK, monoterpenes; carbonyl compounds include acetaldehyde, acetone, butanone, pentanone; alcohol compounds include methanol, ethanol; sulfur compounds include methanethiol, dimethyl sulfide; while methyl nitrite, C₄H₁₀S and C₃H₈OS were considered individually.

Changes in the manuscript:

3.2 Rewetting dynamics

To examine whether the soil physicochemical properties measured at four different sites of the B2 TRF (Table 1) contributed to the soil VOC fluxes over the rain rewet, partial least square regression (PLSR) analysis was conducted. In addition to the soil physicochemical properties reported in Table 2, soil moisture (volumetric water content), soil matric potential (soil water availability to plants) and soil respiration measured from the four sites were also included as predictors in the PLSR analysis. For each site, averaged soil VOC fluxes over the last two days of drought and over the first seven hours of rewet were used for PLSR analysis. The soil fluxes of VOCs from the same compound class and that showed similar rewetting

dynamics were averaged. Therefore, isoprenoid compounds include isoprene, C_5H_8O , MACR+MVK, and monoterpenes; carbonyl compounds include acetaldehyde, acetone, butanone, and pentanone; alcohol compounds include methanol and ethanol; sulfur compounds include methanethiol and dimethyl sulfide; soil fluxes of methyl nitrite, $C_4H_{10}S$ and C_3H_8OS were considered individually. Soil fluxes for each individual VOC from the four sites of the B2 TRF over the last two days of severe drought and over the first seven hours of the rain rewet are shown in Figure S4. In Figure 5a-g regression coefficients and variable of importance (VIP) are shown from each individual PLSR analysis. Soil moisture, soil matric potential, soil respiration, and soil clay content were the most important variables ($VIP > 1$) for the most of the VOCs. Higher soil water content during the rain rewet, induced a higher release in the ambient air of carbonyl compounds (Figure 5b) accumulated in the soil micropores and, at the same time, a higher abiotic dissolution alcohol compounds (Figure 5c) from the ambient air. Additionally, increased soil water content after rewet also increased the microbial activity, leading to a higher soil uptake of isoprenoids (Figure 5a) and to a higher soil emission of sulfur compounds (Figure 5d). In contrast, methyl nitrite and C_3H_8OS soil emissions negatively correlated with soil respiration indicating that soil emissions of these compounds decreased with increasing soil microbial activity. Soil clay content negatively correlated with all soil VOC fluxes. The effect of soil clay content was interconnected to the soil water content. Higher soil water content was associated with lower VOC sorption capacity of the clay minerals due to its hydrophilic character. Moreover, when clay minerals are wetted, they can swell, decreasing porosity, VOC diffusion, and therefore both VOC emission and uptake. This explains the negative effect of soil clay content on soil VOC emissions and uptakes over the rain rewet.

Figure 5 Regression coefficients of partial least squares regression (PLSR) models and Variable Importance (VIP) for the covariance between the measured soil variables at four sites of the Biosphere 2 Tropical Rainforest and a) isoprenoid compounds soil uptake, b) carbonyl compounds soil emission, c) alcohol compounds soil uptake, d) sulfur compounds soil emission, e) methyl nitrite soil emission, f) C_3H_8OS soil emission, and g) $C_4H_{10}S$ emission over the rain rewet. For isoprenoid compounds, soil fluxes of isoprene, C_5H_8O , MACR+MVK, and monoterpenes were averaged; for carbonyl compounds, soil fluxes of acetaldehyde, acetone, butanone, and pentanone were averaged; for alcohol compounds, soil fluxes of methanol, and ethanol were averaged; for sulfur compounds, soil fluxes of methanethiol, and dimethyl sulfide were averaged. Positive regression coefficients indicate a positive relationship and negative ones a negative relationship. Variables with a VIP > 1 are considered important while variables with VIP < 1 are considered less important.

Organic matter is an important VOC adsorber in soil, Organic matter contents? As another important soil adsorber, clay content showed a significant effect on VOCs dynamics in soil even seems higher than moisture content parameters in Fig. 5. And this might also answer partially to statements like line 129

Thanks for your comment. We did not directly measure soil organic matter content. However, since about 58% of the mass of organic matter exists as carbon and that the amount of inorganic carbon was previously measured to be negligible in these soils, the total carbon reported in Table 1 is a good proxy of the soil organic matter.

Reviewer #3 (Remarks to the Author):

I am happy with the revisions and don't have additional comments.

Thank you.